

# Spectral optical properties of soot: laboratory investigation of propane flame particles and their link to composition

Johannes Heuser[1,*], Claudia Di Biagio[2,*], Jerome Yon[3], Mathieu Cazaunau[1], Antonin Bergé[2], Edouard Pangui[1], Marco Zanatta[1,4], Laura Renzi[4], Angela Marinoni[4], Satoshi Inomata[5], Chenjie Yu[2], Vera Bernardoni[6], Servanne Chevaillier[1], Daniel Ferry[7], Paolo Laj[8], Michel Maillé[1], Dario Massabò[9], Federico Mazzei[9], Gael Noyalet[2], Hiroshi Tanimoto[5], Brice Temime-Roussel[10], Roberta Vecchi[6], Virginia Vernocchi[11], Paola Formenti[2], Benedicte Picquet-Varrault[1], and Jean-Francois Doussin[1]

[1] Univ Paris Est Creteil and Université Paris Cité, CNRS, LISA, F-94010 Créteil, France
[2] Université Paris Cité and Univ Paris Est Creteil, CNRS, LISA, F-75013 Paris, France
[3] INSA Rouen Normandie, Univ. Rouen Normandie, CNRS, Normandie Univ., CORIA UMR 6614, 76000, Rouen France
[4] Institute of Atmospheric Sciences and Climate, National Research Council of Italy, Bologna, Italy
[5] NIES, National Institute for Environmental Studies, Tsukuba, Japan
[6] Department of Physics, Università degli Studi di Milano & National Institute of Nuclear Physics INFN-Milan Italy
[7] Aix Marseille Univ., CNRS, CINaM, Marseille, France
[8] Univ. Grenoble Alpes, IRD, CNRS, INRAE, Grenoble INP, IGE, 38000 Grenoble, France
[9] Department of Physics University of Genoa & INFN-Genoa, Italy
[10] Aix Marseille Univ., CNRS, LCE, Marseille, France
[11] National Institute of Nuclear Physics INFN-Genoa, Genoa, via Dodecaneso 33, 16146, Italy

*Correspondence to*: Johannes Heuser (johannes.heuser@lisa.ipsl.fr) and Claudia Di Biagio (claudia.dibiagio@lisa.ipsl.fr)

## Abstract

Soot aerosol generated from the incomplete combustion of biomass and fossil fuels is a major light-absorber, however its spectral optical properties for varying black carbon (BC) and brown carbon (BrC) content remains uncertain. In this study, propane soot aerosols with varying size, maturity, and composition, i.e. elemental to total carbon ratio (EC/TC), have been studied systematically in a large simulation chamber to determine their mass absorption, scattering, and extinction cross sections (MAC, MSC, MEC), single scattering albedo (SSA), and Absorption and Scattering Ångström Exponents (AAE, SAE). Apart from the MSC, all other parameters show a variability associated with the soot EC/TC ratio in soot. The MAC at 550 nm increases for increasing EC/TC, with values of 1.0 $m^2g^{-1}$ for EC/TC=0.0 (BrC-dominated soot) and 4.6 $m^2g^{-1}$ for EC/TC=0.79 (BC-dominated soot). The AAE and SSA at 550 nm decrease from 3.79 and 0.29 (EC/TC=0.0) to 1.27 and 0.10 (EC/TC=0.79). A combination of our results for propane soot with literature data for laboratory flame soot from diverse fuels supports a generalized exponential relationship between particle EC/TC and its MAC and AAE values, representing the spectral absorption of soot with varying maturity to lie in an optical continuum. From this, we extrapolate a MAC of 7.9 and 1.3 $m^2g^{-1}$ (550 nm) and an AAE (375-870 nm) of 1.05 and 4.02 for pure EC (BC-like) and OC (BrC-like) propane soot. The



established relationship can provide a useful parameterization for models to estimate the absorption from combustion aerosols and its BC and BrC contributions.

## 1 Introduction

Carbonaceous aerosol produced during the incomplete combustion of biomass and fossil fuels directly affect the Earth's radiative balance due to the absorption and scattering of atmospheric radiation, making it the largest non-gaseous

contributors to anthropogenic climate forcing (Bond et al., 2013; Jacobson, 2001, 2004; Ramanathan and Carmichael, 2008; Samset et al., 2018; IPCC, 2023). In the climate discussion, absorbing particles produced during the combustion processes are commonly represented by two fractions, namely black carbon (BC) and organic carbon (OC). The term BC is used to refer to a low wavelength-dependent and highly light-absorbing fraction of carbonaceous aerosols and is often synonymously used with elemental carbon (EC, the more thermally stable carbonaceous fraction as obtained from thermo-

optical measurements), refractory BC (rBC, the refractory fraction as obtained from laser incandescence techniques), or equivalent BC (eBC, an optically active absorbing specie from filter-based absorption measurements) (Lack et al., 2014; Petzold et al., 2013). OC is used to refer to the non-refractory/thermally unstable fraction of carbonaceous aerosols, which can contain a large number of organic species of which the light-absorbing ones constitute the so-called Brown Carbon (BrC) (Andreae and Gelencsér, 2006). Differently from BC, the BrC fraction shows a highly wavelength-dependent

absorption, strongly increasing towards the UV (Kirchstetter et al., 2004; Saleh et al., 2013). The fractions EC and OC, plus possibly the inorganic carbon, sum up the total carbon (TC) content of aerosols. Soot is a BC-containing particle formed at high temperatures in flames as an aggregate of nucleated gaseous by-products of incomplete combustion (Haynes and Wagner, 1981; Smith, 1981; Tree and Svensson, 2007). Soot displays a characteristic fractal-like structure (Forrest and Witten, 1979) composed by the aggregation of primary spheres with a diameter of 5 to 40 nm (Dastanpour and Rogak, 2014;

Rissler et al., 2013). Composition-wise soot is characterized by its dominant carbon content which is split into more ordered graphite-like and more disordered molecular structures, the occurrence of which depends on the maturity of the soot (Johansson et al., 2017; Michelsen, 2017). Soot is abundant in urban environments, downwind of wildfires, and in remote areas, and is distributed from the surface to the upper atmospheric layers (Liu et al., 2020a; Pang et al., 2023).

A necessary input to represent soot aerosols and their BC and BrC components in radiative transfer models, both used in

climate simulations and remote sensing retrieval algorithms, is their spectral optical properties. Commonly used parameters for this are the single scattering albedo (SSA; i.e. the ratio of the scattering over extinction) and the mass (absorption, scattering, extinction) cross sections (MAC, MSC, MEC, units of $m^2$ $g^{-1}$), which are used to relate modelled mass concentrations to the aerosol spectral optical properties. The MAC (and similarly the MSC and the MEC) at a specific wavelength is defined as the ratio of the absorption coefficient at that wavelength ($b_{abs}(\lambda)$, units of $m^{-1}$), to the mass

concentration of the specific absorbing aerosol fraction ($m_m$, units of $g$ $m^{-3}$) as:



$$MAC(\lambda) = \frac{b_{abs}(\lambda)}{m_m} \tag{1}$$

Historically, the interest of the scientific community was put into the optical main absorbing fraction, the BC, so that the MAC of soot particles is often related to the MAC of BC, i.e. using $m_{BC}$ in Eq. (1) referring to rBC, EC, or eBC measurements, most commonly determined at wavelengths where the BrC contribution is expected to be negligible (at 550

nm or higher). For freshly emitted BC a commonly accepted value for the MAC is $7.5 \pm 1.2$ m$^2$g$^{-1}$ at 550 nm, as proposed by Bond & Bergstrom (2006), further discussed by Bond et al. (2013), and more recently accepted by Liu et al. (2020b) who proposed a value of $8.0 \pm 0.7$ m$^2$g$^{-1}$ at 550 nm as a representative value for BC-dominated mature fresh soot. In atmospheric measurements, however, the observed MAC values measured for BC-containing aerosols display a high variability with values spanning from 5.5 to 45.9 m$^2$ g$^{-1}$ at 550 nm (Genberg et al., 2013) and from 4.2 to 19.9 m$^2$ g$^{-1}$ at 637 nm (Zanatta et

al., 2016). Very limited knowledge is instead available for the MSC of BC-containing aerosols: only a few estimates report values between 0.92 and 2.12 m$^2$ g$^{-1}$ in the mid-visible (Dastanpour et al., 2017; Schnaiter et al., 2003). The intensity of light scattered compared to light absorbed per mass unit indicates that spectral extinction (MEC) is expected to be dominated by absorption.

As of today, the understanding of variability and range of spectral MAC, MSC, MEC, and SSA of BC-containing aerosols and BC- and BrC-mixtures, such as soot, is still not sufficient enough to adequately represent them in models and remote sensing schemes, resulting in a large spread of utilized input parameters and model assumption. Sand et al. (2021) for example, compare 11 models using MAC values ranging from 3.11 to 17.7 m$^2$g$^{-1}$ for BC and from 0.05 to 0.3 m$^2$g$^{-1}$ for BrC resulting in a large range of estimated aerosol absorption optical depth values for these species. This lack of knowledge on

the optical properties of BC- and BrC-containing aerosols, and the consequent difficulty in representing them in radiative transfer schemes, arises from two main limitations:

1/ *capability to provide artefact-free measurements of absorption and homogeneously-defined cross-sections*. From the point of view of measurements, assessing the absolute value of the MAC, MSC, MEC, and SSA requires minimizing uncertainties linked to the estimation of the optical coefficients and the mass concentration. Measuring the optical properties, in particular

spectral absorption, is challenging for highly absorbing particles, due to measurement artefacts affecting the filter-based instruments, widely used in atmospheric observations (Bond et al., 1999; Collaud Coen et al., 2010; Weingartner et al., 2003; Vecchi et al., 2014). On the other hand, the mass concentration, both for the total aerosols including both BC and BrC ($m_{total}$) and its BC- or BrC-mass fractions only, can be obtained by using diverse proxies ($m_{EC}$, $m_{rBC}$, $m_{eBC}$, $m_{OC}$), techniques and protocols that can rely on detailed characterization of different particles properties, including its composition, morphology,

or effective density (Lack et al., 2014; Petzold et al., 2013; Zanatta et al., 2016). As of today neither standard calibration material nor standard instruments are identified for BC and BrC mass quantification and light–absorption measurements (e.g., Gysel et al., 2011; Baumgardner et al., 2012; Sipkens et al., 2023). The use of the diverse mass concentrations ($m_{total}$, $m_{EC}$, $m_{rBC}$, $m_{eBC}$, $m_{OC}$) in Eq. (1) can lead to different cross section values, therefore artificially spreading the range of





observations. Despite the efforts of field and laboratory communities towards the harmonization of definitions and analysis
protocols, including the work of the Aerosol, Clouds and Trace Gases Research Infrastructure (ACTRIS) community (e.g. Savadkoohi et al., 2023; Cuesta-Mosquera et al., 2021; Müller et al., 2011), the calculation of the MAC, MSC, MEC, and SSA for BC- and BrC-containing aerosols remains heterogeneous in the literature, which limits the capacity of synthesis of available observations.

2/ *parameterization of atmospheric variability including fresh and aged conditions, and linking composition, morphology,*
*and mixing state.* From the point of view of atmospheric variability, the range of observations in the literature, as obtained both from field and laboratory studies, supports the hypothesis that the absorption and scattering properties of soot aerosols vary at the source as a function of the generation conditions, and further modify during lifetime due to atmospheric physico-chemical processing. A variety of different works found that the change of optical properties at the source depends largely on the soot maturity and composition, in particular the organics (BrC-like) and the amount of graphitic fraction (BC-like)
generated (e.g. Cheng et al., 2019; Ess and Vasilatou, 2019; Kelesidis et al., 2021; Malmborg et al., 2021; Schnaiter et al., 2006; Yon et al., 2021) and the size of the single soot particles (Corbin et al., 2022), while the significance of variable morphology is an ongoing discussion (Kahnert and Devasthale, 2011; Liu et al., 2008; Radney et al., 2014). Varying composition, commonly expressed as the EC/TC or EC/OC ratio, and/or varying optical properties have been observed in connection with the combustion of different fuels (e.g. Bescond et al., 2016; Cheng et al., 2019; Colbeck et al., 1997; Olson
et al., 2015) and with variable combustion conditions, in particular, varying ratios of air and fuel expressed as global equivalence ratio or the C/O-ratio (Schnaiter et al., 2006; Mamakos et al., 2013; Moore et al., 2014; Török et al., 2018; Ess and Vasilatou, 2019; Cheng et al., 2019; Ess et al., 2021; Malmborg et al., 2021; Bescond et al., 2016). Observed changes include both the intensity of interaction with light (absolute values of MAC/MSC/MEC/SSA) but also its spectral dependence, represented by the Absorption and Scattering Ångström Exponents (AAE, SAE) (Cheng et al., 2019; Ess et al.,
2021; Ess and Vasilatou, 2019; Kelesidis et al., 2021; Schnaiter et al., 2003, 2006; Török et al., 2018). Further, aging processes, including coagulation, wetting, photo-ageing, and inorganic and organic coating formation, can affect soot properties (e.g. Liu et al., 2022; Hallett et al., 1989; Colbeck et al., 1990; Wang et al., 2017), which may result in changes in the spectral optical properties of soot particles as they age in the atmosphere (Bond et al., 2006; Cappa et al., 2019; Lack and Cappa, 2010; Lefevre et al., 2018; Liu et al., 2015; Schnaiter et al., 2003; Zhang et al., 2008, 2018). Still, assessing the
dependence of the spectral mass cross-sections and SSA at the source due to diverse combustion conditions, and disentangling the changes due to different atmospheric ageing processes, remains difficult and a matter of ongoing research.

In this work, we investigate the variability of spectral MAC, MSC, MEC, and SSA of soot aerosols and the link between spectral optical properties and composition for aerosol generated under different combustion conditions. To do so,
systematic experiments were performed in a large simulation chamber on laboratory-generated flame soot with diverse physico-chemical properties, in particular with diverse EC/TC fractions. Taking advantage of the realism of aerosol suspension in a large chamber and, at the same time, of the control of experimental conditions that laboratory studies



provide, we aim to measure the changes of MAC, MSC, MEC, and SSA of soot with varying BC and BrC content. By merging with past literature studies, additionally, this study tries to identify generalized tendencies describing the cross-section dependencies (absolute value and spectral variability) on soot composition.

## 2 Experimental set-up and instrumentation

Experiments were performed in the 4.2 m$^3$ stainless steel CESAM chamber (Experimental Multiphasic Atmospheric Simulation Chamber; Wang et al., 2011; https://cesam.cnrs.fr/), widely used in past years to investigate the formation processes and the physico-chemical and spectral optical properties of different aerosol types, including primary and secondary species (Caponi et al., 2017; De Haan et al., 2017, 2023; Denjean et al., 2014, 2015; Di Biagio et al., 2014, 2017, 2019). CESAM is a controlled and steady environment designed to allow multiphase atmospheric (photo-) chemistry. Temperature, pressure, and humidity, as well as the gaseous and aerosol content, can be varied in CESAM to reproduce atmospherically-relevant conditions. Realistic solar irradiation is provided by three xenon arc lamps. The homogeneity of the mixture is ensured by a stain-less steel fan placed at the bottom of the reactor which allows a mixing time of around one minute. The particle lifetime in the chamber (i.e. the time necessary to get a particle number concentration reduction by a factor e) was estimated by previous studies to vary from 6 to more than 24 hours for particles between 50 and 200 nm (Lamkaddam et al., 2017), allowing investigation of ageing processes under realistic timeframes. CESAM is a multi-instrumented platform: it is equipped with 12 circular flanges to allow a large panel of instruments to be connected to measure online and offline the thermodynamic state of the volume, its gaseous composition, and aerosol concentration and properties. At the end of each experiment, the CESAM chamber is evacuated to around 3×10$^{-4}$ hPa to avoid carryover contaminations and stays evacuated between subsequent experiments. The chamber is also additionally cleaned manually at the beginning of each campaign.

A total of 22 soot experiments were performed in CESAM to investigate the properties of fresh soot, split into three campaigns that occurred in February–March, May, and October 2021, as summarized in Table S1. A wide range of state-of-the-art online and offline techniques were combined to measure the physical, chemical and spectral optical properties of soot and its BC and BrC components, as well as the gas phase and the chamber's thermodynamic state. A summary of instrumentation together with information on measurement uncertainties is given in Tables S2 and S3. Only the main aerosol measurements used in this study are described in Section 2.3.

## 2.1 Generation of Combustion Soot

The combustion soot aerosol used in the experiments was generated using a diffusion flame soot generator miniCAST (Model 6204 Type C, Jing Ltd., Switzerland). The miniCAST is a reference instrument providing repeatable and controllable soot generation spanning fuel-lean to fuel-rich conditions (Moore et al., 2014). This has been proven to produce soot particles with varying composition, morphological properties, size distributions, and absorption behaviour (Bescond et al.,



2016; Ess et al., 2021; Ess and Vasilatou, 2019; Mamakos et al., 2013; Moore et al., 2014; Schnaiter et al., 2006).
MiniCAST fuel-lean soot is often considered to be comparable to diesel and aircraft (kerosene-type) emissions (Ess et al., 2021; Moore et al., 2014; Saffaripour et al., 2017).

The miniCAST produces air-diluted soot with an $N_2$-quenched co-flow propane-air diffusion flame. The combustion conditions of the miniCAST can be varied by adjusting the mass flow rates of the employed gases, which are the fuel (Propane, Air Liquide, purity >99,95%), the mixing gas, the quench gas (both, evaporated from liquid $N_2$, Messer, purity
>99.995%), the oxidation air, and the dilution air (both, Alphagaz 2 Air, Air Liquide, ≥99,9999%). Computer-controlled mass flowmeters are used to precisely set the flow rates of the different gases in the miniCAST. In the present study, five burning conditions were selected from the predefined miniCAST operation points (OP, from OP1 to OP5) to generate soot with varying properties. The different OP points correspond to constant dilution and quench flows (5 and 2 L min$^{-1}$, respectively) and varying flows for fuel, oxidation air, and mixing $N_2$, resulting in a range of different fuel-to-air ratios
$(m_F/m_A)$. The global equivalence ratio ($\varphi$), defined as

$$\varphi = \frac{(m_F/m_A)}{(m_F/m_A)_{ST}} \qquad (2)$$

with $(m_F/m_A)_{ST}$ the stoichiometric fuel-to-air ratio of complete combustion, varies for the selected points between 0.88 and 1.2. Values $\varphi < 1$ represents fuel-lean combustion (OP1 to OP4) and $\varphi > 1$ represents fuel-rich combustion (OP5). Hereafter the generated soot from the five OP points of the miniCAST will be referred to as CAST soot (CS) and numbered
dependent on the selected OP (CS1 to CS5). A summary of miniCAST operating conditions is reported in Table 1.

**2.2 Experimental protocol**

At the beginning of each experiment, the CESAM chamber was filled with a mixture of 80% $N_2$ (Messer, evaporated from liquid, purity >99.995%), generated by evaporation from liquid nitrogen out of a pressurized liquid nitrogen tank, and 20% bottled $O_2$ (Linde 5.0, purity ≥ 99,999%). The chamber pressure was set at a slight overpressure (+5 hPa compared to
ambient conditions) to avoid external contaminations. Before CS injection, the aerosol background and the instrumental baselines were verified. Aerosol background concentration never exceeded 0.05 μg m$^{-3}$ and 200 particles cm$^{-3}$, corresponding to about 0.4% at the lowest injected CS mass concentration. A particle-free $N_2$ and $O_2$ air flow was continuously injected into CESAM to compensate for the sampling flow rate of the different instruments to maintain stable pressure in the chamber throughout the experiments. The CS was injected in the CESAM chamber by connecting the
miniCAST generator output to the chamber through an activated carbon denuder (details in Supplementary Text S1). Injections lasted from 30 to 390 seconds resulting in a concentration range in CESAM between 14 and 200 μg m$^{-3}$ (as derived from size distribution and effective density measurements, Sect. 2.3.1). The full set of soot data is available from about 10 to 15 minutes after injection.

Once injected in CESAM, the soot aerosols were left in suspension for timeframes varying between 2 and 27 hours in order
to investigate the evolution of particle physico-chemical and optical properties under different controlled aging conditions. In



this study, we consider only measurements done under dry (0% relative humidity) and dark conditions to characterize the soot and to look at the effect of lifetime in a $O_2+N_2$ atmosphere and test the reproducibility of results. All data considered in the present study were acquired at ambient temperature (T = 288-299 K).

Control experiments were performed with submicron ammonium sulphate and sulphuric acid aerosols with the aim of validating the performances and consistency of optical and physico-chemical observations. The results of control experiments, are presented and discussed in the Supplementary Text S2.

### 2.3 Online and offline aerosol measurements

#### 2.3.1 Physico-chemical aerosol properties

The number size distribution between 19.5 and 881.7 nm over 64 channels (dN/dlogD$_m$; D$_m$, mobility diameter) was measured at 3-minute resolution by means of a scanning mobility particle sizer (TSI SMPS) equipped with an [85]Kr neutralizer, a DMA 3080, and a CPC 3772, operated at 2.0/0.2 L min$^{-1}$ sheath/aerosol flow rates. Measurements were corrected for diffusion losses and multiple charge effects with the instrument software. The total particle number concentration was separately measured at 1 second time resolution with a condensation particle counter (CPC TSI 3775, sampling flow rate 1.5 L min$^{-1}$) sensing particles in a diameter range from 4 to larger than 3000 nm.

The measurements of the refractory black carbon particle mass concentration (m$_{rBC}$) and rBC size distribution between 89 and 567 nm (volume equivalent diameter, D$_{ve}$) were performed with a single particle soot photometer (SP2 – Droplet Measurement Technologies, sampling flow rate 0.12 L min$^{-1}$; Stephens et al., 2003) using laser-induced incandescence. The SP2 was calibrated using fullerene and size- and mass-selected CS from the miniCAST (following Laborde et al. (2012)) and analysed assuming a material density of 1.8 g cm$^{-3}$. The measured size distributions were extrapolated based on a lognormal fit in order to approximate the rBC-mass concentrations over the full size range (Liu et al., 2014).

The mass concentration of non-refractory chemical components (m$_{non-refr}$) including nitrate, sulphate, ammonium, chloride, and organics were measured using a time-of-flight aerosol chemical speciation monitor (TOF-ACSM-Aerodyne, sampling flow rate 1 L min$^{-1}$) at 6-min resolution and a vaporizer temperature at around 600°C. The instrument was calibrated with monodispersed ammonium nitrate and ammonium sulphate particles and data were extracted by assuming a collection efficiency of 1.

The total aerosol mass concentration (m$_{total}$) in 5 min steps was measured online by means of a tapered element oscillating microbalance (TEOM 1400a – Thermo Scientific, sampling flow rate 3 L min$^{-1}$) maintained at 300 K. To note that due to the TEOM sensitivity to even slight pressure changes and potential consequential data instability, its data are mainly used to validate mass calculations from complimentary approaches described below.

Additional techniques were used to supplement the continuous online measurements and characterise specific soot physico-chemical properties at selected moments during the experiments. Mass-selected size distribution measurements were performed using an SMPS (TSI, X-Ray neutralizer 3087, DMA 3080, CPC 3775 high) and centrifugal particle mass analyser



(CPMA – Cambustion; Olfert & Collings, 2005) in a DMA-CPMA-CPC setup as described in Yon et al. (2015). For these measurements the CPMA is used to select particles according to their mass-to-charge-ratio, while the DMA-CPC in SMPS

mode measures the number distributions of the CPMA-selected single particle mass ($m_p$). An effective density ($\rho_{eff}$) of the aerosol was calculated for each selected $m_p$ as (Kasper, 1982; Park et al., 2003):

$$\rho_{eff}(D_m) = \frac{6m_p}{\pi D_{m,fit}^3} \tag{3}$$

where $D_{m,fit}$ is the median diameter from the lognormal fit of the measured size distribution of the $m_p$-corresponding SMPS scan. Measurements of $\rho_{eff}$ were performed, if particle abundance allowed it, for 10 values of $m_p$ from 0.03 to 6 femtogram

(fg, $10^{-15}$ g, selected with the CPMA) in order to characterize the relation between $\rho_{eff}$ and $D_m$ over the full soot size range. On average a complete acquisition corresponded to about 35 minutes of measurements including scan and CPMA stabilization periods. The DMA-CPMA-CPC scans were performed after soot injection in the chamber, at selected points during the experiment, and at the end of each experiment. Combining SMPS ($dN/dlogD_m$) online number distribution measurements with $\rho_{eff}(D_m)$ from the measurements allows a second way to retrieve the $m_{total}$ as:

$$m_{total} = \sum_i \frac{\pi}{6} D_{m,i}{}^3 \frac{dN}{dlogD_{m,i}} dlogD_{m,i}\, \rho_{eff}(D_{m,i}) \tag{4}$$

with the subscript i indicating the channels of the SMPS. The $\rho_{eff}(D_{m,i})$ is the effective density extrapolated over range of the SMPS $D_{m,i}$ channels calculated based on the power-law relation of effective density and diameter of the aggregates (Park et al., 2003; Rissler et al., 2013):

$$\rho_{eff}(D_m) = a(D_m)^{D_{fm}-3}. \tag{5}$$

In Eq. (5) $a$ is a constant term and $D_{fm}$ is the mass-mobility exponent, a parameter describing the morphology of the aerosols ($D_{fm}$=3 for spherical particles, around $D_{fm}$=2.2 for soot particles (e.g. Sorensen, 2011; Mamakos et al., 2013; Maricq and Xu, 2004)).

Aerosol filter samples were taken for offline measurements of soot composition, including EC and OC by thermo-optical analysis (Sunset EC/OC analyser, Sunset Laboratory Inc.) using the EUSAAR 2 protocol (Cavalli et al., 2010), and soot

morphology by Transmission Electron Microscopy (TEM) imaging. Soot particles for EC/OC measurements were collected on pre-baked (550°C for 8 hours) quartz-fiber filters (47–mm diameter, Pall TissuquartzTM, 2500 QAT-UP) using a custom-made stainless-steel filter holder. The sampling started when the soot concentration was stable. Particles were sampled at flow rates between 2 and 10 L min⁻¹ for about 60 to 240 min. However, since particle concentrations in the chamber resulted often in collected masses below the quantification limit (QL= 0.42 µg cm⁻²), additional soot sampling was performed directly

out of the miniCAST. In this case, the sampler was directly connected to the output of the soot generator through a charcoal denuder, for which the charcoal sheets were changed after each sampling. This sampling configuration is equivalent to sampling soot directly after the injection into the CESAM chamber. The flow rate used for direct sampling was varied around between 4 and 6.5 L min⁻¹ and the sampling lasted between 6 and 360 sec. For each cast soot, different filter samples were therefore obtained by combining chamber and direct sampling. The ensemble of results was averaged to obtain for each





CS an average and standard deviation of EC/TC, EC/OC, and OC/TC ratios representative of each soot type. Tests were performed by adding a backup filter behind the aerosol filter during the sampling in order to determine the volatile fraction of OC. Results indicate no presence of any notable quantities of volatile fractions. Blank filters were collected during chamber and direct sampling experiments and show OC and EC values below the detection limit (DL= 0.25 μg cm$^{-2}$). In between sampling and analyses all filters were stored in a freezer at −20 °C.

Soot particles for transmission electron microscopy analyses were collected on TEM-grids (200-mesh copper grid with a Formvar/carbon film, Agar Scientific) placed on a 47-mm Teflon filter (Nuclepore, Whatman, 0.8 μm nominal pore size) using a custom-made stainless-steel filter holder. A layout of three grids per filter was used. Sampling was performed at 2-4 L min$^{-1}$. Samples were taken over different moments of the experiments to determine the morphology of soot. Sampling time was set based on number concentration to ensure that similar area loadings would be sampled. The TEM images were

acquired with a JEOL® 100CXII microscope. The pictures were processed to isolate the soot via Fiji (Schindelin et al., 2012) an open-access image processing package of ImageJ (Schneider et al., 2012) using the trainable WEKA segmentation plugin (Arganda-Carreras et al., 2017) with the FastRandomForest classifier. The software was used to generate two class images, namely soot particles and background. Pixel groupings of less than 0.04% of pixels were filtered out as they could be attributed to noise. An automated method based on Euclidean distance mapping (EDM-SBS), described in Bescond et al.

(2014), was applied to these two class images to determine the average diameter of the primary particles composing the soot aggregate ($D_{pp}$) and its fractal dimension ($D_f$). The primary particle size and the fractal dimension enable the approximated description of the soot particles over their self-similar structure as:

$$N = k_f \left( \frac{D_g}{D_{pp}} \right)^{D_f} \tag{6}$$

in which N is the number of spheres in the aggregate and $D_g$ the particles gyration diameter (Sorensen, 2011).

**2.3.2 Spectral aerosol optical properties**

The extinction coefficient ($b_{ext}$) at 450 and 630 nm was measured online at 1 second resolution by two Cavity Attenuated Phase Shift monitors (CAPS - CAPS PM$_{ex}$, Aerodyne Research, sampling flow rate 0.85 l min$^{-1}$, Kebabian et al. (2007); Massoli et al. (2010)). Additionally, in some experiments, two CAPS single scattering albedo monitors (CAPS PM$_{SSA}$, Aerodyne Research, sampling flow rate 0.85 L min$^{-1}$, Onasch et al. (2015)) were deployed to provide further measurements

of extinction at 450 and 630 nm (scattering coefficient data from the CAPS PM$_{SSA}$ were not considered for the present experiments). The $b_{ext}$ by the CAPS PM$_{ex}$ and PM$_{SSA}$ agreed within uncertainty.

The aerosol scattering coefficients ($b_{scat}$) at 450, 550, and 700 nm at angles between 7° and 170° were measured at 1 sec resolution by a nephelometer (TSI Model 3563 Integrating Nephelometer, sampling flow rate 2 L min$^{-1}$, Anderson et al., 1996; Anderson and Ogren, 1998) connected online to the chamber. To account for the limited field of view of the

instrument the truncation correction factor ($C_{trunc}$), i.e., the ratio of the $b_{scat}$ at 0–180° and 7–170°, needs to be estimated. To this aim, the formulation proposed by Massoli et al. (2009) for absorbing sub-μm particles was applied. The Massoli et al.



correction requires as input the real part of the complex refractive index (n) of the aerosol. A default value of 1.95 for n through all wavelengths was assumed for the calculations based on the commonly used value proposed by Bond and Bergstrom (2006). The resulting $C_{trunc}$ varies between 1 and 1.044. Potential artefacts caused by this fixed value of n were

estimated and are considered in the uncertainty of the total scattering signal (see Table S2). The truncation corrected $b_{scat}$ values were used to interpolate the scattering at 630 nm based on the Scattering Ångström Exponent (SAE) calculated as the power-law fit of $b_{scat}$ vs the wavelength ($b_{sca} \sim \lambda^{-SAE}$).

The absorption coefficient ($b_{abs}$) at 450 and 630 nm was derived from online measurements throughout each experiment as the difference between $b_{ext}$ and $b_{sca}$ (extinction-minus-scattering approach, EMS) from the CAPS monitors and the

nephelometer. Three filter-based techniques for absorption measurements were applied in order to complement and validate the $b_{abs}$ retrieved online by EMS: the Multi Angle Absorption Photometer (MAAP; Petzold et al., 2005; Petzold & Schönlinner, 2004), the Multi-Wavelength Absorbance Analyzer (MWAA; Massabò et al., 2013, 2015), and the polar photometer of the University of Milano (PP_UniMI; Bernardoni et al., 2017; Vecchi et al. 2014). The basic principle of these techniques is the same and based on the MAAP concept. The instruments measure both the transmittance through an aerosol

loaded-filter matrix as well as the reflectance at different angles (130° and 165° for the MAAP, 125° and 165° for the MWAA, and between 0 and 170° at an angular resolution of 0.4° for the PP_UniMI). Together with a radiative transfer scheme based on the Mie theory and a two-stream approximation, the measurements of transmitted and reflected light intensity are used to determine the aerosol absorption coefficient. The main advantage of the MAAP, MWAA and PP_UniMI is that $b_{abs}$ is retrieved without the need to correct for the typical artefacts of filter-based techniques like the

aethalometer, as the multiple scattering in the filter fiber and the scattering effect by aerosols are taken into account in their radiative transfer scheme.

The key differences between MAAP, MWAA, and PP_UniMI are the operating wavelengths and the deployment mode, therefore determining a different temporal resolution and set up for CESAM experiments:

- The MAAP is a single-wavelength field–deployable instrument operating at 637 nm and typically 1-min resolution. The

MAAP was connected online to CESAM during experiments. However, because of its high flow rate (typically of 16.7 L min⁻¹, but reduced to 8 L min⁻¹ for the chamber experiments), it was used to measure soot absorption only at specific moments, i.e. shortly after injection in CESAM and towards the end of each experiment, for time intervals ranging between 30 minutes to a maximum of 4 hours.

- The MWAA and the PP_UniMI are laboratory–based setup operating at 5 wavelengths (375, 405, 532, 635, 850 nm for

the MWAA and 405, 448, 532, 635, 780 nm for the PP_UniMI). The MWAA and the PP_UniMI work offline with aerosol collected on filter samples, therefore they integrate observations across a sampling period that can be variable depending on the concentrations (minutes to hours). In the present study, the filter tape from the MAAP (Glass filter fiber GF 10) was cut and removed after each measurement interval in CESAM so that the collected filtered spot could serve as the filter samples for the MWAA and PP_UniMI. Filters were stored in a freezer at -20°C in-between sampling and

measurement. Blank filters were taken to evaluate possible artefacts in the observations.



To note that these filter-based techniques, despite being assumed as a reference, can also be affected by a loading effect artefact, i.e. the accumulation of particles on the filter reduces the linearity in the retrieved absorption signal, therefore causing an underestimate of the absorption coefficient for increasing aerosol deposition. For the MAAP, as detailed in (Hyvärinen et al., 2013), this happens when the accumulation rate is higher than 0.04 µg min$^{-1}$. In this study, when relevant,

the $b_{abs}$ from the MAAP was corrected for the loading effect following the formulation proposed by (Hyvärinen et al., 2013) based on the use of the measured raw reflected signal at 165°. No correction schemes for possible loading effects are available for the MWAA and the PP_UniMI. Previous intercomparison test between these instruments and Hyvärinen-corrected MAAP data showed that when the absorbance of the filter sample (ABS, i.e. the optical depth of the particle-loaded filter) is ABS < 1 no need for further corrections exists.

Together with the MAAP, MWAA, and PP_UniMI, filter–based measurements of spectral absorption was performed by aethalometers connected to CESAM online throughout experiments, including both a single-spot AE31 model (Weingartner et al., 2003) and a dual-spot AE33 (Drinovec et al., 2015). Due to the high aerosol loading effects during soot experiments, as well as the uncertainty on the multiple scattering corrections, data from these instruments are not used in the present analysis. However, data from present experiments will be used to study the multiple scattering corrections and performances

of the aethalometer for highly absorbing aerosols (Renzi et al., in preparation).

In addition to chamber measurements on the total aerosol fraction, measurements of the spectral absorption for CS1 to CS5 were also performed in the 190 to 640 nm range at 2 nm resolution on soot soluble extracts. The aim of these additional measurements is to determine the signature of the soluble BrC in order to better separate the contribution of BC and BrC to the total absorption. For these analyses, the CS aerosols were sampled directly out of the miniCAST on 47 mm pre-baked

quartz-fiber filters (Pall Tissuquartz TM, 2500 QAT-UP) and extracted in acetonitrile (Fujifilm Wako pure chemical). The extracts were analysed using a combination of high performance liquid chromatography (HPLC; Agilent 1260 infinity) for separation of different constituents, a UV-Vis photodiode array detector (PDA) measuring the spectral absorbance of compounds eluting from the HPLC column, and an electrospray ionization high-resolution mass spectrometer (ESI/HRMS; JEOL JMS-T100LP AccuTOF LC-plus). The absorbance in these measurements is defined as $\log\left(\frac{I_0}{I}\right)$ where $I_0$ and $I$ are the

intensity of the incident and transmitted light. The PDA absorbance spectra, integrated across elution time, are the only measurements used in the present work. Details on experimental measurements and signal treatment are provided in Text S3.



# 3 Data analysis



**Figure 1** **Timeline of a typical CS experiment. The plot shows (from top to bottom): a) the contour plot of the number size distribution in mobility diameter (dN/dlogDm) obtained from the SMPS. The number median diameter (NMD) obtained from lognormal fitting of the size distribution is overplotted (black line); b) the mass concentration ($m_{total}$) obtained from SMPS and $\rho_{eff}(D_m)$ data based on Eq. (4) and directly measured by TEOM microbalance; c) the extinction and scattering coefficients ($b_{ext}$, $b_{sca}$) at 450 and 630 nm measured by CAPS-PM$_{ex}$ and the nephelometer (extrapolated based on the SAE); d) the single scattering albedo (SSA) at 450 and 630 derived from $b_{ext}$ and $b_{sca}$ measurements and the SAE (from $b_{sca}$ 3-λ data); e) the calculated MAC and f) MSC and MEC at 450 and 630 nm. The data is at a 3-minute resolution with exception of the TEOM that is at a 5-minute resolution. Data reported in this figure correspond to the experiment CS1 performed on 19/10/2021.**





## 3.1 Data homogenisation

An example of a timeline for a typical experiment analysed in this work is shown in Fig. 1. Data acquired during the experiments from different instruments were first treated to homogenize their temporal resolution. Data were taken from

about 15 min after the injection in CESAM. For all online techniques, measuring at different time resolutions, the data were averaged over common base time intervals of 3 minutes corresponding to the time resolution of the SMPS used to derive the total aerosol mass concentration. Furthermore, all volumetric quantities were converted into standard pressure and temperature (STP) conditions assuming 1013.25 mbar and 273.15 K and considering the measured temperature and pressure for each instrument acquisition.

**3.2 Data validation of total aerosol mass concentration and absorption coefficient**

By combining the number distribution and the $\rho_{eff}(D_m)$ data for each CS, an estimate of the total soot aerosol mass concentration $m_{tot}$ was retrieved (Eq. 4) along each experiment, as exemplary illustrated in Fig. 1b. The SMPS-CPMA retrieved $m_{tot}$ shows good agreement with online measurements of the TEOM microbalance under consideration of their associated uncertainties.

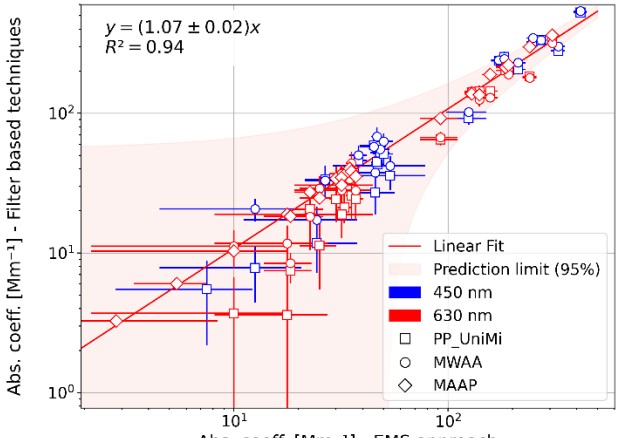


**Figure 2 Absorption coefficient calculated at 450 and 630 nm with the extinction-minus-scattering method (EMS) compared to the filter-based values of the PP_UniMI, MWAA, and the MAAP for all CESAM experiments on CS. For comparison purposes, the EMS and MAAP online measurements are averaged over the PP_UniMI and MWAA sampling times (15 to 120 min for the different experiments). The $b_{abs}$ values from the MWAA are interpolated at 450 based on the Absorption Ångström Exponent**
**(AAE) calculated as the power-law fit of $b_{abs}$ vs the wavelength ($b_{abs}\sim\lambda^{-AAE}$) while the PP_UniMI original data at 448 nm are taken. For comparison at 630 nm, the MAAP data at 637 nm and the MWAA and PP_UniMI data at 635 nm are considered. A linear fit of the relation between the measurements of the two stiles was calculated with an assumed intercept at zero and added in red together with a 95% confidence interval of the fit. The correlation coefficient ($R^2$) for the fit is also indicated.**

The extinction, scattering and absorption coefficients ($b_{ext}$, $b_{scat}$, $b_{abs}$) were measured with a 3-minutes resolution over the
aerosol lifetime inside the chamber, as exemplary illustrated in Fig. 1 (c). The comparison of the absorption coefficients measured with the EMS method (filter-free) compared to the filter-based techniques (MAAP, MWAA, PP_UniMI) at 450





and 630 nm for the ensemble of CS experiments is shown in Fig. 2. The comparison indicates an excellent correlation ($R^2$=0.94) and an average 7% difference between EMS and filter-based techniques, well within measurement uncertainties, therefore supporting the consistency of the different absorption measurements. Higher data dispersion is observed at

absorption intensities below 10 $Mm^{-1}$, where lower values of the absorption are measured by the offline MWAA and PP_UniMI compared to the online MAAP and the EMS technique. This can be explained as a combination of low loading and limited sensitivity for low soot concentrations for the offline techniques. Data below 10 $Mm^{-1}$ from MWAA and PP_UniMI were removed from any further analyses.

**3.3 Retrieval of aerosol mass cross sections at 450 and 630 nm**

The MAC, MSC, and MEC were calculated at 450 and 630 nm at 3-minute resolution for each experiment based on Eq. (1) considering the $b_{abs}$ (from EMS), $b_{sca}$ (from nephelometer), and $b_{ext}$ (from CAPS $PM_{ex}$) and using the total aerosol mass ($m_{total}$) from SMPS (($dN/dlogD_m$) and $\rho_{eff}(D_m)$ (Eq. 4; Fig. 1 (e-g)). The $m_{total}$ was used for the cross section calculations with the aim of describing the properties of the entire soot particles and taking both absorbing fractions of BC and BrC into account. The $\rho_{eff}(D_m)$ used in the $m_{total}$ calculation is the value measured in correspondence to each CS experiment. When a

dedicated $\rho_{eff}(D_m)$ measurement was not performed for the specific experiment, the average and standard deviation of the $\rho_{eff}(D_m)$ for the closest condition was used. The uncertainty on the 3-minute derived MAC, MSC, and MEC was estimated to average at 12%, 14%, and 11%, respectively, as calculated from the statistical error propagation formula based on the uncertainty of optical coefficients and mass concentration. The ensemble of the 3-min resolution MAC, MSC, and MEC of all experiments were combined for each CS point to get a best-guess statistical synthesis of those parameters and their

uncertainty at 450 and 630 nm. For this, each measured value with its mean measurement uncertainty was represented as a Gaussian distribution using:

$$f(x) = \frac{c}{\sigma\sqrt{2\pi}} e^{-\frac{1}{2}*\left(\frac{x-\mu}{\sigma}\right)^2} \tag{7}$$

where $\mu$ is the measured value, $\sigma$ the mean measurement uncertainty of $\mu$, and $c$ the number of times the mass cross section value was measured normalized by the total measurements for the CS point. Then, the sum of all MAC, MSC, and MEC

probability distribution functions was calculated and fitted with a Gaussian distribution. The $\mu$ and $\sigma$ from the summed Gaussian function are the resulting best-guess estimate and 1-$\sigma$ uncertainty of the MAC, MSC, and MEC for each CS condition. An illustrative example of the statistical analysis performed on cross section measurements is shown in Fig. S1.

**3.4 Retrieval of the aerosol absorption and scattering Ångström exponent and the cross sections at 550 nm**

For each CS point, an average value of the AAE between 375 and 870 nm was derived as the power-law fit of the multi-

wavelength MAC estimated over the visible range by combining all available $b_{abs}$ datasets (from the MWAA, PP_UniMI, EMS, and MAAP) normalized using $m_{total}$. The AAE was then used to calculate the MAC at 550 nm from the value at 450 nm as $MAC_{550}=MAC_{450} (450/550)^{AAE}$.



The AAE of the acetonitrile-soluble soot fraction (AAE $BrC_{ACN}$) was calculated for each CS point as the power-law fit between the integrated absorbance and the wavelength, measured from HPLC-UVVIS within the two spectral ranges 300-
450 nm and 370-450 nm. The ranges are limited by the intensity of the absorbance which prohibits the extension of the calculation above 450 nm.

For each CS point the SAE was derived as the average of SAE values obtained from the power-law fit of the $b_{scat}$ vs λ from nephelometer observations over 3-minute resolution. The values were averaged for all relevant fresh soot conditions on the same principle as the MAC (Eq. (7)) assuming a normal distribution of the SAE values to consider both statistical variability
as well as the measurement uncertainty. The SAE was used to calculate the MSC at 550 nm from the value at 450 nm as $MSC_{550} = MSC_{450} (450/550)^{SAE}$. The MEC at 550 nm was obtained as the sum of $MAC_{550}$ and $MSC_{550}$, with an average uncertainty of around 19%

### 3.5 Retrieval of the aerosol single scattering albedo (SSA)

The SSA at 450 and 630 nm was calculated at 3-minute resolution as the ratio of $b_{sca}$ and $b_{ext}$ along the experiments (Fig.
1(d)). The CS average values at 450, 550, and 630 nm were estimated as MSC/(MAC+MSC) taking the statistical synthesis values for each soot. The propagated uncertainty on the SSA varies between 12 and 25%.

## 4 Results

A summary of miniCAST operating conditions and average CS physico-chemical properties is reported in Table 1.

### 4.1 Carbonaceous aerosol composition

The composition of the CS varies between the combustion conditions. The soot OC fraction increases and the particle size decreases with the reduction of the oxidation air and the addition of the mixing air from CS1 to CS5, suggesting the generation of more mature (CS1) to less mature (CS5) soot, in agreement with observations with other miniCAST models (i.e., Török et al., 2018; Schnaiter et al., 2006; Ess and Vasilatou, 2019). The EC/TC is fairly similar (0.79, 0.73) for CS1 and CS2, reducing to 0.67 and 0.53 for CS3 and CS4, respectively. The fuel-rich condition of CS5 produces soot dominated
by OC, with EC concentrations under the quantification limit for thermo-optical measurements, resulting in an EC/TC ratio that therefore we assume to be equal to 0.0. This is in agreement with SP2 measurements indicating the presence of a small but not quantifiable rBC fraction for CS5. In contrast, the ACSM detected a significantly lower amount of organic aerosol compared to thermo-optical analyses (going from around 1.5% of the total soot mass for CS1 to about 2.3% for CS5; not shown). The low organic content in the ACSM can be due to the particle bouncing off from the vaporizer and/or can be
linked to the nature of the organic material itself (Mamakos et al., 2013; Maricq, 2014; Török et al., 2018). Which is in line with the observation that a significant fraction of the OC detected in thermo-optical analyses is associated with pyrolytic carbon.





**Table 1** Physico-chemical properties of the Cast Soot (CS) aerosols for the five combustion conditions considered in this study
(CS1 to CS5) corresponding to the first five predefined miniCAST operation points (OP1 to OP5). For each CS, the table reports:
the flowrates of the different gases used in the miniCAST with constant dilution and quench flows (5 and 2 l min-1, respectively);
the combustion condition (fuel lean/rich); the global equivalence ratio ($\varphi$) estimated from Eq. (2); the count median diameter
(CMD, $D_m$) estimated from the lognormal fitting of the SMPS number distribution for soot after 60 min of injection in CESAM
(with their standard deviation); the mobility exponent ($D_{fm}$) retrieved from fitting effective density measurements as in Eq. (5); the
fractal dimension ($D_f$) and primary particle diameter ($D_{pp}$) retrieved from TEM images analysis; the elemental versus total carbon
ratio (EC/TC) as obtained from thermo-optical analyses. For TEM-retrieved parameters note that an extended statistic is
available for CS1 (11 sets of 100 images from different experiments and individually analysed, providing 11 estimates of $D_f$ and
$D_{pp}$, here reported as averages and stdev). Conversely, for CS2 to CS4 the statistic is reduced (1 set of 100 images analysed for each
CS point) and for CS5 the number of images was not enough to retrieve reliable information on $D_f$ and $D_{pp}$, which are not reported
for this CS5.

| Cast Soot Type | miniCAST flowrates fuel/mix. N₂/ox. Air [L min⁻¹] | Combustion condition | Global equivalence ratio ($\varphi$) | CMD₆₀ₘᵢₙ ($D_m$) [nm] | Mobility exponent ($D_{fm}$) | Fractal dimension ($D_f$) | Avg. Primary particle diameter ($D_{pp}$) [nm] | EC/TC |
|---|---|---|---|---|---|---|---|---|
| CS1 (OP1) | 0.03/0/0.75 | Fuel lean | 0.91 | 145 ± 12 | 2.11 ± 0.04 | 1.91 ± 0.22 | 9.8 ± 1.9 | 0.79 ± 0.11 |
| CS2 (OP2) | 0.025/0/0.60 | Fuel lean | 0.95 | 138 ± 1 | 2.10 ± 0.04 | 2.04 ± 0.24 | 7.2 ± 2.5 | 0.73 ± 0.08 |
| CS3 (OP3) | 0.025/0.01/0.60 | Fuel lean | 0.95 | 122 ± 9 | 2.10 ± 0.04 | 1.79 ± 0.20 | 15.4 ± 1.9 | 0.67 ± 0.09 |
| CS4 (OP4) | 0.023/0.02/0.60 | Fuel lean | 0.88 | 103 ± 17 | 2.20 ± 0.04 | 1.83 ± 0.22 | 10.0 ± 1.7 | 0.53 ± 0.13 |
| CS5 (OP5) | 0.023/0.02/0.45 | Fuel rich | 1.20 | 79 ± 2 | 2.25 ± 0.04 | – | – | 0.00 ± 0.22 |

## 4.2 Size distribution

The size distributions of the generated aerosols show specific features for the different CS and they evolve during chamber
experiments, as illustrated in Fig 1a. Figure 3 shows a comparison of the number size distribution for the five CS for the first
SMPS scan after injection in CESAM (Fig. 3a) and 60 min later (Fig. 3b). For the fresh soot right after injection in CESAM
(Fig. 3a), the size distributions of all EC-dominated soots (CS1 to CS4) show bimodal log-normal distributions. They display
a main peak between roughly 95 to 120 nm and an additional smaller peak ranging from 25 to 40 nm. The organic-
dominated CS5 on the other hand is the only mono-modal aerosol in the first scan displaying a peak at around 40 to 45 nm. It
has to be noted, that the number of particles smaller than 50 nm is increasing from CS1 to CS5, following the decrease in
fuel and oxidation air. The lifetime of the small peak is rather short and the mode is only distinctively identifiable for the
first few minutes of the chamber lifetime as the particles coagulate rapidly to form within around 10 to 30 min a mono-
modal size distribution. At 60 minutes after generation (Fig. 3b), all soots from CS1 to CS5 show mono-modal size
distributions spanning a range of median diameters varying between 130 to 150 nm for EC-rich CS1 and CS2, 95 to 125 nm
OC-richer CS3 and CS4 and 75 to 80 nm for the organic-dominated CS5, which is also showing a narrow size distribution.



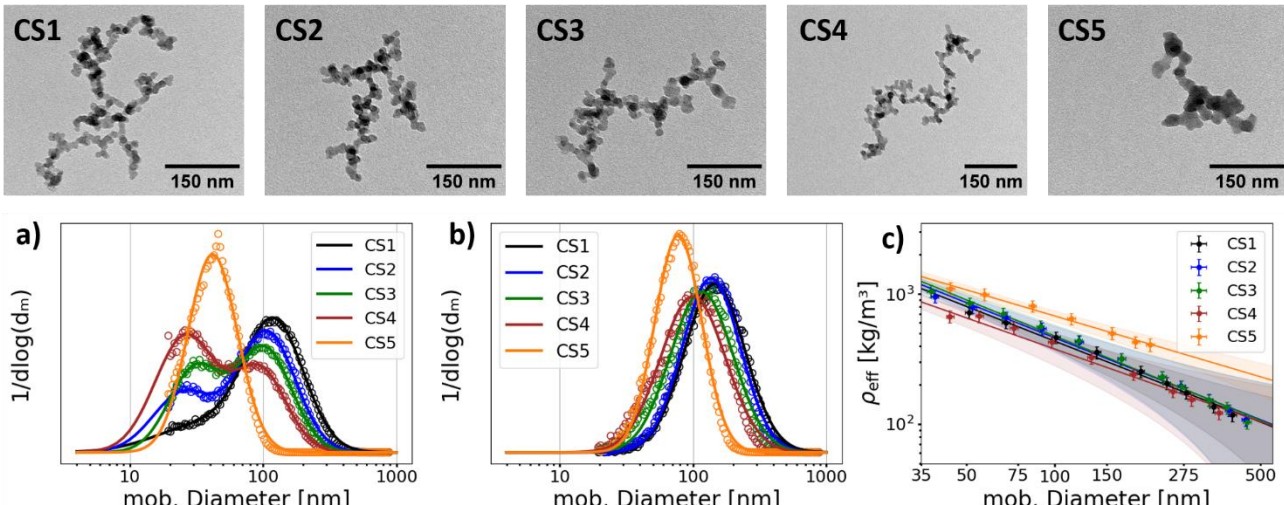

**Figure 3 Top panel shows pictures illustrating the morphology of the five CS as obtained from TEM-imaging for samples collected 20 to 45 minutes after injection in CESAM. Bottom panel: number (dN/dlogD$_m$) size distribution measured (open circles) and fit-extrapolated log-normal size distribution (lines) of generated CS just after completion of the injection in CESAM (a) and 60 minutes later (b). Size data are normalized by the total number for each distribution. (c) Effective density ($\rho_{eff}$) as a function of mobility diameter (D$_m$) for the five CS as measured between a few minutes and 30 minutes after injection. The power-law fit (Eq. 5) is also plotted with its fitting uncertainty (shaded area).**

### 4.3 Morphology

Pictures of the five CS as obtained from TEM-imaging for aerosol sampled about 20-45 min after injection are shown in Fig. 3. The different CS shows the typical fractal aggregate structure of soots and a lacey morphology. While CS1 to CS4 do not show directly visible structural differences, CS5 displays a lower number of large aggregates as well as more rounded/collapsed structures. Formal statistical analysis of TEM pictures confirms limited variability for the fractal dimension (D$_f$) and primary particle diameter (D$_{pp}$) for CS1 to CS4 (D$_f$ = 1.79 - 2.04, D$_{pp}$ = 7.2 - 15.4 nm), conversely, no information is available for CS5 due to a lack of statistics. For CS1 the range of measured D$_{pp}$ and D$_f$ are in the range of values observed for CS2 to CS4, suggesting that the variability of D$_{pp}$ and D$_f$ for the different CS is within the range of measurements variability of a single CS. Therefore, the TEM measurements do not allow for differentiation of the CS1 to CS4, which have to be considered similar in particle size and aggregate fractal dimension. For CS1-CS4 an average size of around 10 nm for D$_{pp}$ is proposed based on the results of all performed measurements. This diameter sits at the lower end of the primary spheres observed in combustion aerosol (Bescond et al., 2016; Hu and Koylu, 2004; Köylü and Faeth, 1992; Lee et al., 2002; Mamakos et al., 2013). Similarly, a fractal dimension D$_f$ of around 1.86 is proposed for the ensemble of CS1 to CS4. This value is located towards the upper end of observed values (Bescond et al., 2016; Köylü and Faeth, 1992; Wang et al., 2017) and the Diffusion Limited Cluster Aggregates (DLCA) regime which with a fractal dimension of about 1.75-1.8 is considered to represent soot well (Sorensen, 2011).



**4.4 Effective density**

The measured effective density $\rho_{eff}(D_m)$ (Fig. 3c), taken 15 to 45 minutes after soot injection into CESAM, decreases with size as measured for $D_m$ between 40 and 450 nm. The $\rho_{eff}$ varies in the range 0.95 - 0.12 g cm$^{-3}$ for CS1 to CS4 and 1.22 and 0.2 g cm$^{-3}$ for CS5, as retrieved from fitting curves (Eq. 5). The fractal mobility exponent ($D_{fm}$) is 2.10 for CS1-CS3, increasing to 2.20 and 2.25 for CS4 and CS5, a range of values that following Sorensen (2011) is typical for the DLCA soot. The $D_{fm}$ in our study is in line with previous studies on propane miniCAST and ethylene soot (Ess et al., 2021; Mamakos et al., 2013; Maricq and Xu, 2004; Moore et al., 2014), while it is on the low end for values observed for diesel and aircraft turbine engine soot (Durdina et al., 2014; Maricq and Xu, 2004; Olfert et al., 2007; Park et al., 2003; Rissler et al., 2013). The slight differences in $D_{fm}$ indicate a slight change in the morphology between the CSs, the observed higher values of $\rho_{eff}$ and $D_{fm}$ for the fuel-rich CS5 suggest a change in composition and slightly more compact morphology (less elongated, more rounded aggregates) for these soots, in agreement with TEM images. Knowledge of the primary particle size together with the effective density measurement enable the estimation of a density for the primary particles following Yon et al. (2015). For CS1, for which the most robust statistics from TEM images is available, an average density for the primary particles of 1.73 ± 0.14 g cm$^{-3}$ is retrieved, close to the value of 1.8 g cm$^{-3}$ reported and associated with BC in literature (Bond and Bergstrom, 2006; Park et al., 2004; Wu et al., 1997).

**4.5 Spectral optical properties of generated soot**

Measured cross sections and SSA values are found to be stable throughout the aerosol lifetime (varying between 2 and 27 hours) in the isolated CESAM volume, and to be independent of injected mass/number concentrations (Fig. 1e-g). The only exception is found for a slight increase of the values in the first hour of measurements that stabilized around 60 to 90 minutes after soot injection. These changes are generally less than 10% and considered not significant. Stability in values is also observed in the further measurements over the aerosol lifetime, in dark and dry (0% relative humidity) conditions, for the SAE (Fig. 1d) as well as the effective density (not shown), supporting the conclusion that lifetime (over 24h) and coagulation have negligible impact on the soot aerosol chemical, morphological and optical properties.

The results of the statistical analysis taking into account the ensemble of CS experiments for MAC, MSC, and MEC at 450 nm are shown in Fig. 4 (values at a wavelength of 630 are shown in Fig. S2) and summarized in Table 2. As illustrated in Fig. 4, the measured cross section values are generally well represented by normal distributions, despite observed deviations especially in the MSC, likely associated to limited statistics. The fitting parameters indicate that the relative standard deviation determined for the cross sections is fairly similar to the relative measurement uncertainty indicating good repeatability in the combustion aerosol generation and measurements. For all generated soots the MAC is dominating over the MSC and the SSA is within 0.12-0.29 at 450 nm and 0.07-0.27 at 630 nm for the different CS. Retrieved MSC values seem rather independent of CS type and decreasing with wavelength, spanning the range 0.7-0.9 and 0.2-0.4 m$^2$g$^{-1}$ at 450 and 630 nm. The SAE increases from 2.43 for CS1 to 3.81 for CS5. The MAC decreases in the order of CS2, CS1, CS3, CS4 to



CS5 from 6.3 to 2.2 m²g⁻¹ at 450 nm, and from 4.5 to 0.8 m²g⁻¹ at 630 nm. The MEC absolute value and variability mainly

follow MAC tendencies, with values in the range 7.2-3.1 m²g⁻¹ (450 nm) and 4.8-1.1 m²g⁻¹ (630 nm). Marked differences in

MAC and MEC are identified when comparing the fuel-rich CS5 against the fuel-lean CS1-CS4 showing a narrower range of

variability.

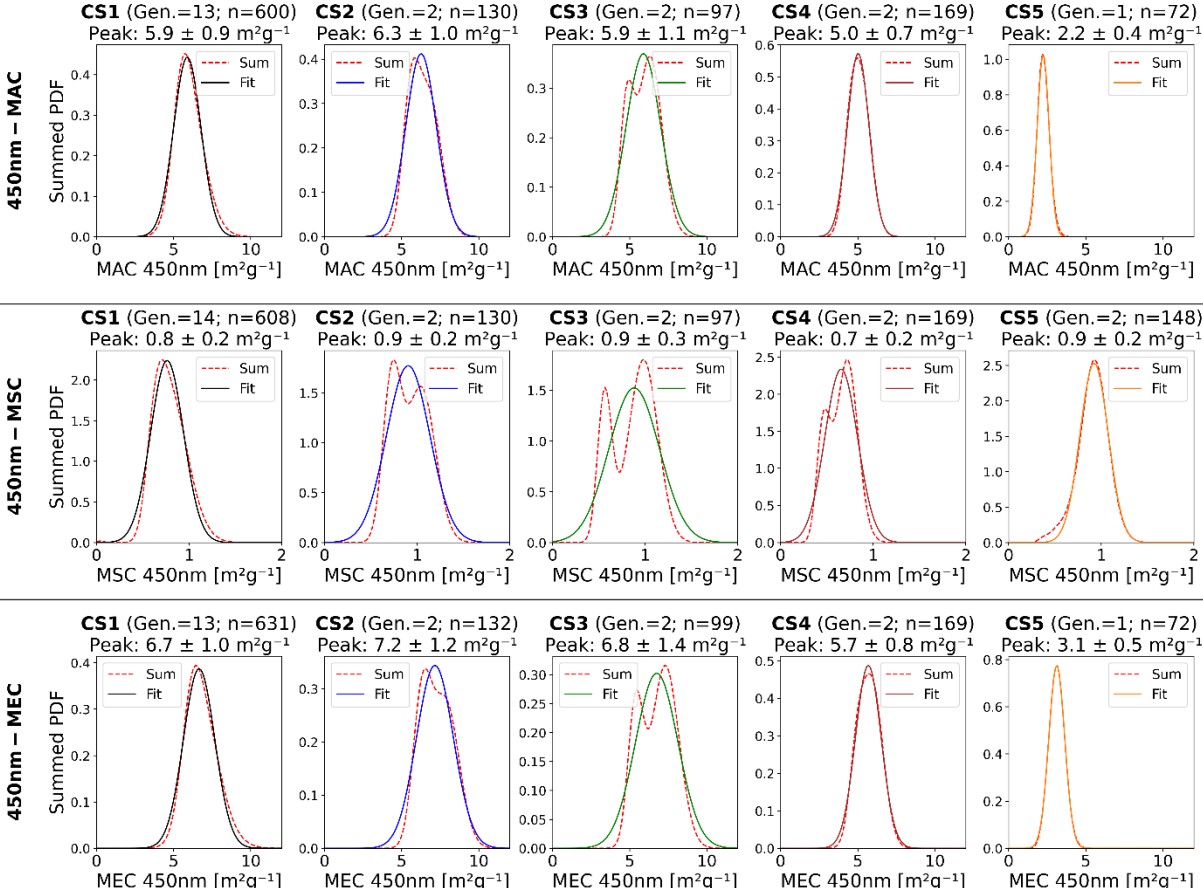

**Figure 4 The sum of Gaussians of MAC/MSC/MEC values and fitted Gaussians distribution of the sum for CS1 to CS5 at 450 nm.**
**For each plot the soot, together with their number of generations, and used data points (n) is given. Further, the mean value from the fit together with its standard deviation is provided, determined from the fit of Eq. (7).**

The wavelength-dependent MAC values obtained for the different CS from combining all experimental data from both EMS and filter-based techniques, and the power law fitting to retrieve the AAE values are shown in Fig. 5. The retrieved AAE is 1.26 for CS1 and increases to 1.34, 1.59, 1.88 and 3.8 for CS2 to CS5, respectively, indicating stronger wavelength

dependence of absorption going from EC-rich to OC-rich soot particles. The increasing wavelength-dependency of absorption for increasing OC content going from CS1 to CS5 suggests the BrC nature of the organic material composing the soot.





**Table 2 Mean of the MAC, MSC, MEC and SSA estimated from normal distribution summation and fitting at 450 and 630 nm for the five cast soot (CS1-CS5) and extrapolated at 550 from 450 nm based on the AAE. The mean of the AAE and SAE is also indicated. Provided uncertainties represent the combination of statistical and measurement uncertainties for the MAC, MSC, MEC, and SSA parameters, while for the AAE and SAE uncertainties are related to the fitting procedure.**

| Cast Soot Type | MAC (m² g⁻¹) | | | MSC (m² g⁻¹) | | | MEC (m² g⁻¹) | | | SSA | | | AAE | SAE |
|---|---|---|---|---|---|---|---|---|---|---|---|---|---|---|
| | 450 nm | 550 nm | 630 nm | 450 nm | 550 nm | 630 nm | 450 nm | 550 nm | 630 nm | 450 nm | 550 nm | 630 nm | (375-850 nm) | (450-700 nm) |
| CS1 | 5.9 ± 0.9 | 4.6 ± 0.8 | 4.3 ± 0.7 | 0.8 ± 0.2 | 0.5 ± 0.1 | 0.3 ± 0.1 | 6.7 ± 1.0 | 5.1 ± 0.8 | 4.6 ± 0.8 | 0.12 ± 0.03 | 0.10 ± 0.03 | 0.07 ± 0.02 | 1.27 ± 0.13 | 2.43 ± 0.18 |
| CS2 | 6.3 ± 1.0 | 4.8 ± 1.1 | 4.5 ± 0.6 | 0.9 ± 0.2 | 0.5 ± 0.1 | 0.4 ± 0.1 | 7.2 ± 1.2 | 5.3 ± 1.1 | 4.8 ± 0.7 | 0.13 ± 0.02 | 0.10 ± 0.03 | 0.08 ± 0.02 | 1.36 ± 0.21 | 2.54 ± 0.15 |
| CS3 | 5.9 ± 1.1 | 4.3 ± 1.0 | 4.1 ± 0.6 | 0.9 ± 0.3 | 0.5 ± 0.2 | 0.4 ± 0.1 | 6.8 ± 1.4 | 4.8 ± 1.0 | 4.5 ± 0.6 | 0.13 ± 0.04 | 0.10 ± 0.04 | 0.09 ± 0.03 | 1.59 ± 0.22 | 2.72 ± 0.17 |
| CS4 | 5.0 ± 0.7 | 3.4 ± 0.7 | 3.6 ± 0.4 | 0.7 ± 0.2 | 0.4 ± 0.1 | 0.2 ± 0.1 | 5.7 ± 0.8 | 3.8 ± 0.8 | 3.8 ± 0.5 | 0.12 ± 0.03 | 0.11 ± 0.04 | 0.05 ± 0.02 | 1.88 ± 0.31 | 2.82 ± 0.34 |
| CS5 | 2.2 ± 0.4 | 1.0 ± 0.2 | 0.8 ± 0.3 | 0.9 ± 0.2 | 0.4 ± 0.1 | 0.3 ± 0.1 | 3.1 ± 0.5 | 1.4 ± 0.2 | 1.1 ± 0.3 | 0.29 ± 0.05 | 0.29 ± 0.09 | 0.27 ± 0.09 | 3.79 ± 0.33 | 3.81 ± 0.33 |

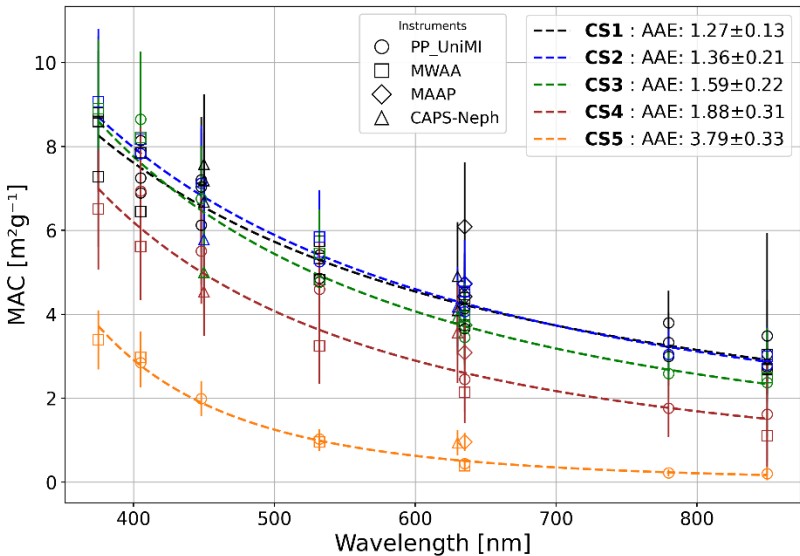

**Figure 5 Wavelength-dependent MAC values from combining the data from both filter-based (MWAA, MAAP, PP_UniMI) and EMS techniques for fresh soot CS experiments. As for Fig. 2 the EMS and MAAP data are averaged over the same intervals of MWAA and PP_UniMI filter-based measurements. Vertical error bars represent the cross section uncertainties consisting of both the uncertainty of the measurement of mass and absorption coefficient. The latter being the measurement uncertainties for the offline techniques and for the online techniques (MAAP and EMS) the averaged measurement uncertainty over the filter sampling**
**period. The power-law fit line (MAC~λ⁻ᴬᴬᴱ) and the retrieved AAE values (± fit uncertainty) for each CS are indicated. To note that the offline techniques proved sensitive to low filter sample loadings and were not able to measure significant absorption values at high wavelength (870 nm) for CS3-CS5.**



## 4.6 Spectral absorbance of soot soluble extracts

Extractive measurements are used to provide independent evaluation of the absorbing capacity of the soot organic material, further extending the spectral range down to 200 nm. Figure 6 (left panel) shows the absorbance spectra measured for acetonitrile extracts for CS1 to CS5 soot, where data are normalized at the peak to allow a direct spectra comparison between the CS points. The absorbance decreases steadily with wavelength, showing a similar behaviour for CS1 to CS4, and noisy data above 450 nm. A significantly higher absorbance signal in the whole spectral range up to 600 nm is obtained for the OC-dominated CS5. The AAE estimated from 300 to 450 nm (AAE $BrC_{ACN}$ (300−450)) and from 370 to 450 nm (AAE $BrC_{ACN}$ (370−450)) are shown in Fig. 6 (right panel) in comparison to the AAE obtained in Fig. 5 for the bulk soot. The AAE $BrC_{ACN}$ (300−450) and AAE $BrC_{ACN}$ (370−450) values range from 4.8 to 6.0, indicating a strong wavelength dependence of absorption in the 300–450 nm range for the acetonitrile extracts. The AAE $BrC_{ACN}$ does not show a clear dependence on the CS type, i.e. on sampled combustion conditions.

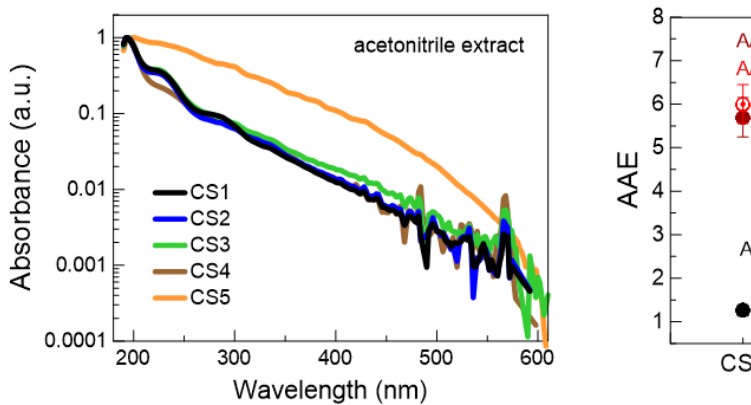
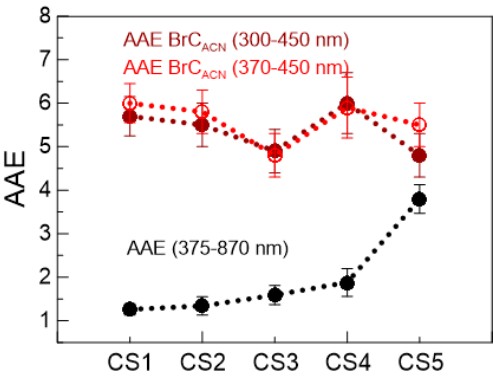

**Figure 6 (Left panel) Wavelength-dependent absorbance for acetonitrile soluble CS1-CS5 soot fraction. Absorbance data are normalised at the peak. (Right panel) Comparison of the AAE obtained in the 300-450 nm and 370-450 ranges for the acetonitrile extracts and the AAE obtained for the soot aerosol within the 375-870 nm range.**

## 5 Discussion

### 5.1 Soot composition and combustion conditions

The composition and size distribution of the CS vary between the generation conditions. It can be observed that the thermo-optical organic carbon is increasing going from fuel-lean to fuel-rich conditions. This increase of organics is observed to correlate well with the median diameter of the aerosol and in particular with an increase of particles in the size range smaller than 50 nm directly after injection that coagulates to form a mono-modal aerosol over particle lifetime. Different publications on miniCAST soot (e.g., Schnaiter et al., 2006; Moore et al., 2014) report similar bi-modal size distributions and attribute the smaller mode to spherical condensed organics from Polycyclic Aromatic Hydrocarbons (PAH) which are



associated with lower absorption and higher wavelength dependency. These PAH-nucleation mode particles coagulate over their lifetime with themselves and with soot aggregates forming larger aggregates. This description fits with the online measurement of the SMPS in our experiments and could explain the evolution of the optical properties. The TEM and effective density measurements were performed after coagulation in the mono-modal state, therefore, the information on the

properties of the small model is extremely limited for our experiments. However, the analysis of TEM pictures shows that all particles consistently present the soot typical agglomerate morphology and does not provide sufficient evidence for the presence of a large number of spherical particles in the size of the nucleation mode (30 nm) and larger, as observed in Schnaiter et al. (2006). Manual analysis, performed as validation of the EDM-SBS-based automated analysis, seems to indicate the potential presence of larger primary particles around the lower range of the nucleation mode especially for CS5

but lacks statistical significance, repeatability, and precision. These observations, of the morphology, in combination with the thermo-optical analysis and the ACSM measurements, show that the semi-volatile organic fraction is increasing for the different operation points but the majority of the OC is only volatilized at high temperatures or is pyrolytic carbon, and do not indicate the presence of high amounts of condensed organic species. These observations correspond to the results of Mamakos et al. (2013), Maricq (2014), and Török et al. (2018) that indicate that the mini-CAST soot contains a significant

amount of refractory organic carbon that is not volatilized by a catalytic stripper, a thermodenuder, or a furnace and shows as pyrolytic carbon in the thermo-optical analysis. They propose that this fraction could be identified as less mature soot. At the same time, extractive measurements performed in the present study show that the organic material, or at least a fraction of it, is soluble in common solvents (i.e., acetonitrile) and confirm its absorbing nature, particularly evident for the OC-rich CS5.

Based on our results and the above-mentioned works, we consider the aerosols from CS1 to CS5 to increasingly contain such

a less mature fraction of soot particles with higher fractions of OC than EC, impacting the composition and size, and also the spectral optical properties of the aerosol. As a matter of fact, the coagulated mono-modal batch aerosol can be considered a mixture of two groups of combustion particulate matter containing soot with different maturity (EC-rich and OC-rich), both light-absorbing. Changing the proportion of these different fractions would explain the differences between the EC-rich mature CS1 and the OC-rich less mature CS5 that is more or less purely organic and made up, based on this interpretation, of

a large fraction of less mature soot and a condensed semi-volatile organic fraction.

Mature EC-dominated soot, as the CS1 in the present study, are found in diesel and aircraft (kerosene-type) emission in the real atmosphere (Ess et al., 2021; Moore et al., 2014; Saffaripour et al., 2017), while less-mature OC-richer combustion particles, as the CS5, are observed with low temperature diesel combustion (Jung and Bae, 2015; E et al., 2022) and the combustion of low quality fuels (Corbin et al., 2019).

## 5.2 Relating the physico-chemical and spectral optical properties of soot

The optical properties of soot are found to be influenced by the combustion condition. While the MSC values do not vary significantly between CS, the MAC values, and therefore also the MEC values, as well as the spectral dependence of





absorption (and scattering), vary strongly between fuel-lean conditions (CS1 to CS4) and the fuel-rich condition (CS5), whereas there are limited differences between CS1, CS2, CS3 and CS4.

The spectral dependence of the absorption expressed as the AAE follows the evolution of the absolute value of fuel, oxidation air, and mixing $N_2$ in the same way the size, the relative presence of the organic fraction, and thus also the presence of the potentially less mature soot fraction does. The AAE slowly increased from close to 1 for the EC-rich/mature dominated CS1, the value generally associated with pure BC (Bond and Bergstrom, 2006; Moosmüller et al., 2011), up to an AAE close to 4 for the dominantly organic CS5, a value that falls into the range more commonly associated with

moderately-absorbing BrC (Lack and Cappa, 2010; Moosmüller et al., 2011; Saleh, 2020). This indicates different optical properties in the differently matured soots and likely varying amounts of condensed volatile organic compounds, in agreement with the literature (Mamakos et al., 2013; Maricq, 2014; Török et al., 2018; Malmborg et al., 2021).

Based on these observations we investigated the relationship between the AAE and EC/TC ratio, similar to Malmborg et al.
(2021). In order to supplement our work with additional observations and also compensate for the lack of values for EC/TC-ratios between 0.1 and 0.5, we added data from various studies in the literature that generated soot using propane miniCASTs, similar combustion systems using ethylene, and from diesel and kerosene experiments (Ess et al., 2021; Moore et al., 2014; Saffaripour et al., 2017). In particular, the work of Ess & Vasilatou (2019) which tested different miniCAST combustion conditions for the miniCAST type 5201 Type BC, provided an extensive set of soot measurements with AAE
and EC/TC-ratios. Other literature studies, such as the one by Cheng et al. (2019), provided AAE values against EC/OC ratios. In this case, the EC/OC ratios were transformed into EC/TC based on the relationship EC+OC=TC and the assumption that the absolute majority of the soot is carbonaceous allowing us to use TC as an approximation of the total aerosol mass. Figure 7a shows the AAE values of the soots from our work and literature plotted against their EC/TC ratio. The AAE values cover a seemingly continuous spectrum of EC/TC ratios and show a continuous increase from about 1 (and
even lower) for high EC/TC values to values of 4 and higher at low EC content. These results agree well with observations by Malmborg et al. (2021) and Cheng et al. (2019) (the latter added in grey in Fig. 7), who observed a power law relation between the AAE and EC/refractory carbon ratio for diesel and miniCAST aerosol and the EC/OC-values for aerosols from the combustion of toluene and benzene, respectively. The key difference between our observations and the ones by Cheng et al. (2019) is found at really low EC-content and which is not available in Malmborg et al. (2021). For values of
EC/TC<20%, the values observed by Cheng et al. (2019) follow an asymptotic power-law behaviour and continue to increase at high organic carbon content, while our values approach a maximal value, for CS5, with an EC-content of effectively zero and are thus not described well by a true power-law. While the power law fit is also able to describe our data under the consideration of the uncertainties, we propose as an alternative an approximation of the trends using the exponential function $AAE = a + b * e^{(-c*EC/TC)}$. The fit of this function displayed in Fig. 7a is only performed on the
propane-combustion-like soot and thus does not include the values by Cheng et al. (2019). This exponential formulation allows to extrapolate the AAE values for pure EC and OC, which would be 1.05 ± 0.28 (EC/TC=1) and 4.02 ± 0.29



(EC/TC=0) for propane-combustion-like soot based on the results in Fig. 7a. The AAE estimated from the independent extractive measurements (AAE BrC$_{ACN}$ in the range 4.8 to 6 ($\pm$0.5)) is higher but comparable within uncertainties with the estimated value for pure OC based on the relationship against the particle EC/TC ratio (4.02).

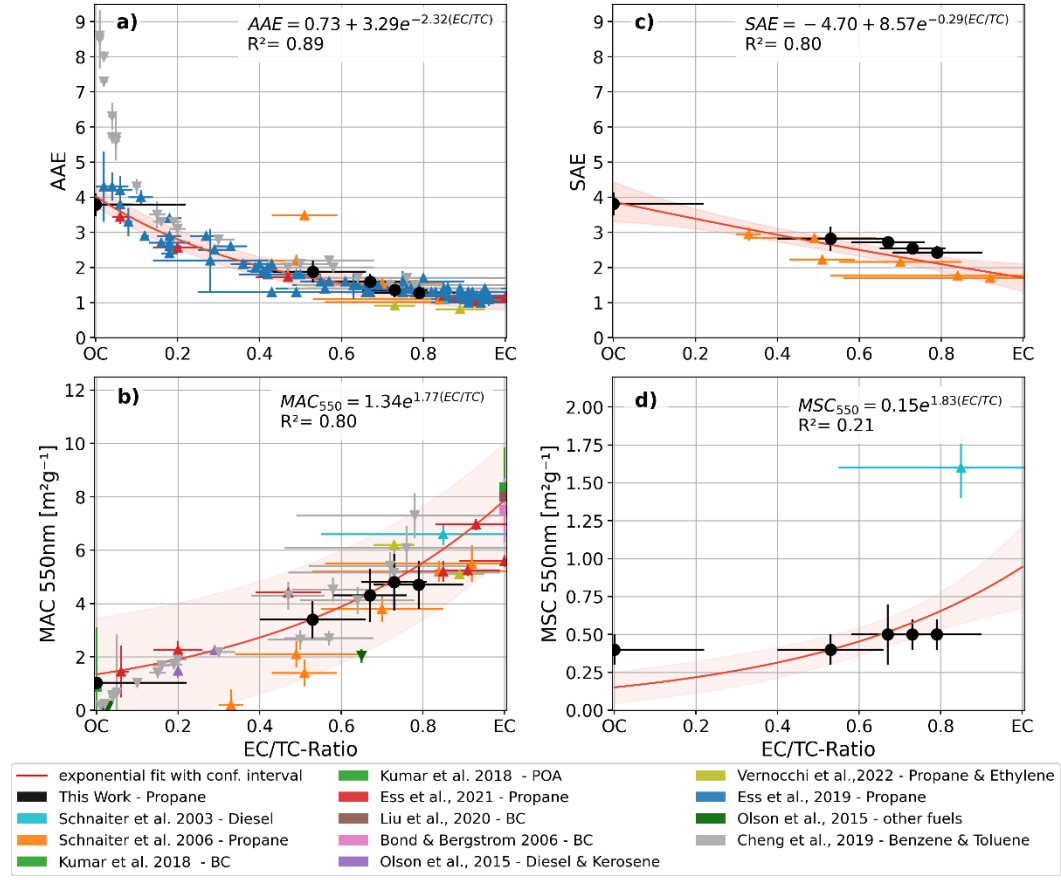

**Figure 7 Compositional dependent absorption mass absorption and scattering cross sections (MAC, MSC) and their spectral variability represented by the absorption and scattering Ångström exponents (AAE, SAE). The MAC data obtained in the present study at 550 nm (black points) are combined with observations from past literature studies and put in relation to the EC/TC soot content. Data are fitted with the equation: $y = a + b * e^{(c*EC/TC)}$. Due to their fuel type, the soot produced by Cheng et al. (2019) (grey) and the other fuels from Olson et al. (2015), as well as values for pure BC and POA (Bond and Bergstrom, 2006; Kumar et al., 2018; Liu et al., 2020b), are not considered in the fits. In order to account for error bars in the fit an arbitrary minimal error of $10^{-3}$ was applied for EC/TC-ratios when these were not provided in their respective source studies. Any other uncertainty in the retrieved values was propagated according to the used calculation. The resulting equations for the absorbing properties are AAE = $(0.73 \pm 0.12) + (3.29 \pm 0.12) e^{-(2.32\pm0.30)\left(\frac{EC}{TC}\right)}$ and $MAC_{550} = (1.34 \pm 0.05) e^{(1.77\pm0.10)\left(\frac{EC}{TC}\right)}$. Uncertainties for the scattering fits are significantly higher and provided in the supplements in Table S4.**

Cheng et al. (2019) further propose a relationship between the MAC and composition, expressed via the EC/OC ratio, and so do other works (e.g. Olson et al., 2015; Bocchicchio et al., 2022). The same EC/TC plot as for the AAE was tested for the MAC at 550 nm. Figure 7b shows the EC/TC-resolved MAC values at 550 nm supplemented by additional MAC data points from propane, diesel, and kerosene soots from the literature, similarly as done for Fig. 7a. Literature data includes MAC



values for both total particle populations and size-selected measurement (e.g. Ess et al., 2021). The values from works not supplying the 550 nm values were extrapolated using the AAE via $MAC_{550} = MAC_\lambda * (\frac{\lambda}{550})^{AAE}$ (where $\lambda$ is the closest wavelength measured to 550 nm). If the AAE was not directly supplied in the literature works, but multiple MAC values at different wavelengths were provided, those were used to inter- or extrapolate the MAC at 550 nm. The MAC reported in the literature, expressed alternatively as $MAC_{EC}$, $MAC_{rBC}$ or $MAC_{BC}$, were transformed into $MAC_{total}/MAC_{TC}$ (i.e. against total

soot mass as in our calculations) using the approximations EC=rBC=BC, EC+OC=TC, and TC=Total mass. The retrieved data points in Fig. 7b again were complemented by MAC data from "other fuels" including the toluene and benzene from Cheng et al. (2019), but also values from Olson et al. (2015) who burned additionally to kerosene and propane also leaf litter, peat, wood pellets, and coal. Data points for pure BC (assuming EC/TC=1), as well as MAC values proposed for pure primary organic aerosol (POA, assuming OC/TC=1), were added to allow a comparison of the extreme values in the full

range of potential values. Similar to the "other fuels" these are not considered when fitting MAC vs EC/TC. Supplemented with these literature values, we can observe a seemingly continuous spectrum of the MAC values as a function of the EC/TC ratio. While the MAC values show a larger scattering (compared to the AAE), most data points are well described by an exponential function. Thus we propose that the relationship between MAC and EC/TC ratio is well described by $MAC = a * e^{(b*EC/TC)}$. The fit of this function again permits the extrapolation of the MAC values for pure EC and OC, which would

be $7.9 \pm 2.17$ m$^2$ g$^{-1}$ (EC/TC=1) and $1.3 \pm 2.15$ m$^2$ g$^{-1}$ (EC/TC=0) at 550 nm, respectively. The extrapolated MAC value is in line with the one proposed by Bond and Bergstrom (2006) ($7.5 \pm 1.3$ m$^2$ g$^{-1}$) for pure BC, and with the more recent literature synthesis by Liu et al. (2020b) ($8.0 \pm 0.7$ m$^2$ g$^{-1}$) for mature EC-dominated soot. The global models used in AeroCom have an average MAC of 8 m$^2$g$^{-1}$ at 550 nm and assume an AAE of 1 (Samset et al., 2018; Sand et al., 2021), values that we confirm for pure BC particles from our analysis.

Similar to the observation on the AAE, the MAC values of alternative fuels (Cheng et al., 2019; Olson et al., 2015) show different behaviour for low EC-content. While the MAC values of propane combustion aerosol (this work; Ess et al., 2021), remain high at low EC/TC values, the MAC values from the "other fuels" decrease towards zero with decreasing EC/TC values. We conclude that we describe, as Cheng et al. (2019), a relationship between the spectral optical properties (AAE and MAC/MEC) that shows that the optical properties of soot can be displayed on a continuum. However, the agreement

between the miniCAST soot, the ethylene, kerosene, and diesel soot, and soots from additional sources only works well for EC/TC larger than 10%. On the other hand, in cases where the organic fraction dominates the total carbon (OC/TC>90%), and potentially also dominates the absorption, we observed differences between the optical properties from propane-generated soot and soot from what we categorized as "other fuels". This is either caused by measurement uncertainties/precision or is indicating a potential dependence on the fuel type, affecting the nature of the BrC fraction.

Similar plots and analyses were made for the MSC and the spectral dependence of the scattering expressed as SAE (Fig. 7c, 7d). The lack of literature data for these values however did not allow to acquire enough information for any significant synthesis. However, while the MSC data do not show a clear dependence, the plot for the SAE suggests a potential similar



relation between composition and spectral dependence of the scattering, as observed for the absorption. Plots relating the optical properties to the also commonly used EC/OC-ratio can be found in the Supplement (Fig. S3).

## 6 Conclusions

In this study, we investigated the physico-chemical and spectral optical properties of soot aerosols based on systematic experiments in the controlled environment of the CESAM simulation chamber. A propane diffusion flame was used to generate soot particles under varying combustion conditions, with a miniCAST generator. We produced five types of fractal soot aerosols with different degrees of maturity, composed of varying mixtures of EC (BC-like) and OC (BrC-like) and diverse size distributions. The soot organic fraction was found to be soluble in common organic solvents, to be identified mostly as pyrolytic carbon in thermo-optical measurements, and to be non-volatile at temperatures below 600 K, as derived from ACSM measurements. Chemical and morphological analyses in the present study suggest that the organic material in soot is a combination of refractory organic carbon and condensed semi-volatile organic fraction.

The dependence of the MAC, MSC, MEC, and SSA for the five generated soot aerosols and their spectral variability, represented by the AAE and SAE, on the soot composition were analysed. While MSC appears independent of the soot EC/TC content (0.4-0.5 $m^2g^{-1}$ at 550 nm), all other parameters show a variability associated with the soot carbon composition. The MAC at 550 nm increases for increasing EC/TC, with values of 1.0 $m^2g^{-1}$ for EC/TC=0.0 and 4.8 $m^2g^{-1}$ for EC/TC=0.79. The AAE decreases from 3.80 for EC/TC=0.0 to 1.26 for EC/TC=0.79. The SSA at 550 nm varies between 0.1 and 0.29 for decreasing EC/TC. By combining the results from this study for propane soot with literature data for laboratory flame soot using diverse fuels, we identified a generalized exponential relationship between the soot's MAC, its spectral variability AAE and the soot's EC/TC content, well representing the variability of absorption in particular for EC/TC >15-20%. We support therefore the interpretation that the MAC and AAE of soot aerosols lie in an optical continuum, which mirrors the continuum of EC/TC, or continuum of graphitization for soot of varying maturity.

From the exponential fitting, we extrapolate a MAC value of 7.9 $m^2 g^{-1}$ at 550 nm and AAE (375-870 nm) of 1.05 for pure EC (BC-like), and a MAC of 1.3 $m^2g^{-1}$ and an AAE of 4.02 for pure OC (BrC-like) propane soot. The generalized relations between MAC and AAE and EC/TC can provide a useful tool for models, where EC is generally used as a proxy for BC, while EC/TC can be traced from emission inventories. The relationship between EC/TC and soot absolute absorption efficiency and spectral variability, as provided in this analysis, is also important for remote sensing applications, in particular, to develop techniques to differentiate the BC-like and BrC-like absorbing components in smoke and pollution plumes. The possibility to have access to compositional information from remote sensing observations is a key development to support the representation of absorbing aerosols components in models.



**Open Research / Data availability**

The retrieved MAC, MSC, MEC and SSA from this study are available within the Library of Advanced Data Products
(LADP) of the EUROCHAMP Data Centre (https://data.eurochamp.org/data-access/optical-properties/, Heuser et al., 2024a,
b, c, d). The CESAM data used in this study are immediately available upon request to the contact author and will also be
made available through the Database of Atmospheric Simulation Chamber Studies (DASCS) of the EUROCHAMP Data
Centre (https://data.eurochamp.org/data-access/chamber-experiments/).

**Competing interests**

The authors declare that they have no competing interests.

**Author contributions**

CDB and JFD conceived the study. JH, CDB, and JFD designed the experiments and discussed the results. JH conducted the
experiments with contributions by CDB, MC, AB, EP, JY, AM, LR, MZ, CY, PF, BPV, and JFD. DM, FM, and VV
performed the MWAA measurements. RV and VB performed the PP_UniMI measurements. SC and GN performed the
thermo–optical measurements. MM and DF performed transmission microscopy measurements. CDB and SI performed the
extractive measurements. HT contributed to funding acquisition and supervision for extractive measurements. JH performed
the full data analysis under the supervision of CDB and JFD and with contributions from JY, MZ, CY, LR, AM, MC, DF,
DM, VV, and VB. BPV, as PI of the CESAM facility, contributed to full experiment realization and management. PL
provided the SP2 used in the experiments. BTR and AM provided the MAAP used in the experiments. CDB provided
funding acquisition and project administration. JH, CDB and JFD wrote the manuscript. All authors reviewed and
commented on the paper.

**Acknowledgements**

The CNRS-INSU is gratefully acknowledged for supporting the CESAM chamber as a national facility included in the
ACTRIS-France Research Infrastructure as well as AERIS (https://www.aeris-data.fr/) for curing and distributing the data
through EUROCHAMP Data Center (https://data.eurochamp.org/). P. Ginot is acknowledged for logistic support with SP2
operations. Contributions by P. Ausset, M. Essani, P. Prati, and M. Hayashi to TEM analyses, MWAA and extractive
measurements are gratefully acknowledged.



**Funding**

This study has been supported by the French National Research Agency (ANR) through the B2C project (contract ANR-19-CE01-0024-01), by the French National program LEFE-CHAT (Les Enveloppes Fluides et l'Environnement – Chimie Atmosphérique) through the BACON project. This work has received funding from the TNA activity of the European Union's Horizon 2020 research and innovation programme through the EUROCHAMP-2020 Infrastructure Activity under grant agreement No 730997. It has been also supported by DIM Qi² and Paris Ile-de-France Region. This work has been performed with the support of the Canon Fundation through a Canon Fundation Fellowship to C. Di Biagio. L. Renzi and M. Zanatta were partially supported by ITINERIS project (IR0000032), the Italian Integrated Environmental Research Infrastructures System (D.D. n. 130/2022 - CUP B53C22002150006) Funded by EU - Next Generation EU PNRR- Mission 4 "Education and Research"

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
