# Peer review of "Spectral optical properties of soot: laboratory investigation of propane flame particles and their link to composition"

_EGUsphere, 2024_

## Author Comment (AC1)

At first, we would like to thank the reviewers for having carefully read the paper and for having provided valuable comments that helped to improve the quality of the manuscript. We have taken into consideration all the comments raised by the reviewers, and changed the paper accordingly. The details of our changes are highlighted in the text. The point by point answers to Reviewer #1 and #2 are provided in the following in blue.

Before addressing the comments of each reviewer, we want to address general changes to the manuscript based on similar remarks made by both reviewers. Remarks were made by the two reviewers towards the general objective and strategy of this study, therefore the second half of the introduction was adjusted to better contextualise the objective of this work and the chosen experimental strategies. A key part of this is an improved description for the reasoning of studying un-collapsed fresh combustion aerosol which is addressed additionally in the answers to each reviewer below.

General feedback by both reviewers indicated the impression of an extensive amount of information outside of the key sections (notably the discussion) of this work, resulting in a too long manuscript. Based on this review, the Sections 2 and 3 were adjusted and shorten. This was done by reducing the information on detailed technical aspects of the experiments and the measurement techniques and moving extensive details, in particular on supplementary techniques, into the supplements of this work. The authors also tried to shorten the text wherever it seemed that redundant information were provided.

Another change in the structure of the manuscript is the removal of the subsections in the Results part. In order to make the text more concise, we grouped the sections describing the soot size distribution, composition, morphology and effective density in a unique paragraph entitled "4.1 Physico-chemical properties of generated soot". We hope this change could help in smoothing and easing the reading of the paper.

We want to additionally point out that the dataset and doi creation through the Database of Atmospheric Simulation Chamber Studies (DASCS) of the EUROCHAMP Data Centre (https://data.eurochamp.org/data-access/chamber-experiments/) for all the experiments used in this work is ongoing and expected to be finalized in the next weeks. We will add the list of doi as Appendix Table A1 in the paper. We have added a preview of Table A1 in the revised version of the manuscript.

**Reviewer No. 1:**

**General comments**

The optical properties of atmospheric soot aerosols are crucial for understanding the role they play in the climate system. Soot has been deemed a vital part of the climate system because of its ability to affect the vertical temperature structure of the atmosphere, affect cloud lifetime, and how soot aerosols can reduce the albedo of e.g. snow once they are removed from the atmosphere through deposition.

The manuscript describes results from well controlled chamber experiments on soot aerosol particles generated with a miniCAST burner during an extensive campaign using the CESAM chamber. The chamber allows for long experiments due to its 4.2 m2 volume. The study describes data obtained using five different setpoints of the miniCAST burner to produce soot aerosol particles (cast soot, abbreviated as CS in the manuscript) of different chemical composition and physical size. The soot aerosols range from mostly elemental carbon (CS1) to mostly organic carbon (OC) in the CS5 experiment.

The data analysis is for the most part sound and appears to be well executed. The figures are clear and informative.

My main concern is whether these experiments are relevant for ambient aerosols since all the experiments (CS1-CS5) show soot that is still in it's fractal state. A fractal state is expected since the experiments were conducted under dry and dark conditions on fresh soot. In the atmosphere, soot cores will collapse if they are hydrophilic, and that will change the aerosols optical properties (see. e.g. https://doi.org/10.1029/2005JD006389). Collapsed fractal aggregates will have very different optical properties (see e.g. https://doi.org/10.1029/1998JD100069) and thus (MAC, MSC, AAE, and SSA). The question remains, how well these CS1-CS5 cases represent ambient aerosol particles and what conclusions can be drawn from these experiments that are relevant for our understanding of ambient aerosols or to be utilized in models.

The shortcoming of the manuscript becomes evident in the introduction when the authors states that the study "...aim to measure..." and "tries to identify generalized tendencies ... on soot composition.". First, vague wordings do not invite further reading. Second, and more importantly, in the introduction it is not stated what is new to science in this study, which would raise the reader's interest. I suggest the authors make a greater effort to highlight what added information comes out from these experiments and how it helps to increase our understanding about atmospheric aerosols. The results based on Figure 7 should be put as a goal of this study and be mentioned in the introduction.

Yes, atmospheric soot are modified and undergo ageing, in particular coating formations leading to a collapses in structure. This results in aged soot often to be more spherical, effectively smaller and to show higher fractal dimensions at constant soot/BC mass, i.e. changing mass cross sections. However the optical properties of fresh un-collapsed soot as measured in this study remains key. Not only do they serve as basis for the measurement and modelling of fresh combustion aerosol (i.e. at emission sites) but also the modelling of soot after processing. This is because many modelling approaches tend to use the fresh aerosol properties and considered the ageing by adjusting these properties according to modelled processing. One common way in which this is done is the use of an enhancement factor to adjust the $MAC_{BC}$ of fresh soot, for collapse and coating. Therefore, to be able to model soot both in fresh and aged conditions soot properties prior to processing and the effect of the processing itself need to be quantified, i.e. optical properties and the magnitude of the enhancement factor.

While this study does not supply parameterisations for the impact of processing, it was performed to provide data for the description of the soot's optical properties prior to processing i.e. fresh combustion aerosol. The result of this study is a systematic description of the mixing of EC (or BC) and OC (or BrC) components at emission and the effect of the mixing of the two on the spectral optical properties of soot aerosols. Conditions are chosen to include likely

ambient soot properties but allow for a general relationship between composition/maturity of soot and its optical properties.

In order to clarify the scope of the paper, the introduction and the conclusion have been modified. In particular, the text has been changed to clarify why the chosen conditions were used and how the work can contribute to improve the understanding of atmospheric combustion aerosol.

In order to reply reviewer's comment, expressions like "...aim to measure..." and "tries to identify generalized tendencies ... on soot composition." Were systematically adjusted to be less vague.

I suggest changing the manuscript's focus on what you know about the aerosols you have generated, thus leaving the speculation related to CS5 to a separate section of the manuscript. This would make for easier and more interesting reading. As it reads now, much of the results are diluted by speculation about the CS5 aerosols. Speculation about CS5 draws too much attention away from the well-defined experiments of CS1-CS4. There is also a mention of an ACSM being present during the experiments, but the data is never properly used. If you are able to extend the analysis to provide more information about the CS5 aerosol particles, and reduce the amount of speculation, please do so.

The feedback was taken into consideration and the balance between the information on CS5 and the other CS was evaluated and adjusted throughout the text, in particular in the Results and Discussion sections. A characterisation of CS5 and in particular the nature of OC for CS5, but also the other CSs, remains however important. Information based on the observation and measurement results or derived from these were kept in this work to characterise CS5 and the OC and constrain its properties. Concerning the use of ACSM data, as discussed in the manuscript, this instrument had only a limited applicability to soot particles. From Setion 4.1 "To note that ACSM detected a significantly lower amount of organic aerosol compared to thermo-optical analyses (going from around 1.5 % of the total soot mass for CS1 to about 2.3 % for CS5; not shown). The low organic content in the ACSM can be due to the particle bouncing off from the vaporizer and/or can be linked to the nature of the organic material itself (Mamakos et al., 2013; Maricq, 2014; Török et al., 2018)"

I like the work you have done relating to Figure 7a and 7b where you have put your research in context with previous research on the topic and included other types of soot aerosol than the miniCAST soot. When reading the manuscript, it becomes clear that the authors have a vast knowledge about previous literature on the subject. In Fig. 7, the authors have made good use of this knowledge. I would like to encourage the authors to do so more often. As it reads now, the interpretation of the results in relation to other studies is often left to the reader, with a list of articles to read and interpret for themselves. I've listed a few examples below:

P17 L491-493: "The diameter sits at the lower end of the primary spheres observed in combustion aerosol (5 refereces)" It would be much more valuable to have the authors analyze the result than give a list of articles.

P17 L494-495: "This value is located towards the upper end of observed values (3 references)" What is the range in those articles?

P18 502-504: "The Dfm is our study is in line with previous studies on propane miniCAST and ethylene soot (4 references), while it is on the low end for values observed for diesel and aircraft soot (5 references)" Can the authors put some numbers on these differences and similarities so that the reader won't have to read all the listed publications for themselves. That would give much added value to your work than to just cite others.

*We thank the reviewer for pointing out at this aspect. Based on this feedback the values in such instances were added to allow the reader to follow easier the contextualisation of the results.*

I would further suggest the authors try to explain what these experiments mean for our understanding of ambient aerosols. The authors have already stated that in the introduction of the manuscript in terms of MAC, MSC, MEC, and SSA of black carbon (BC) containing aerosol particles; i.e. how can your findings be utilized in models or measurements of atmospheric aerosols. I have made a few suggestions further on.

The aerosols, as the authors correctly state, are representative of freshly emitted aerosols. The conditions in the chamber where dark and dry, which is not a condition that one is likely to come across in the atmosphere, at least considering the longer chamber residence times in the chamber which were up to 27 h. It is not clear to me why these conditions were chosen although I realize that these conditions make for an easier data analysis and well constrained conditions. I would like to see the motivation for this and discussion about how the aerosols you have generated can be useful for understanding soot in ambient conditions.

My suggestion is that you include discussion on what happens when these fractal like soot aggregates enter the atmosphere and that your values and findings apply to legacy soot up to a certain point, before they collapse into more compacted shapes due to atmospheric processing, after which the optical properties you report no longer holds true. My suggestion is that you include discussion on at which stage the soot cores will collapse; see https://doi.org/10.1016/j.carbon.2024.119197 and https://doi.org/10.1038/ngeo2901 and references therein. Then you could state that your experiments can be used in models up to a certain point (i.e. before they collapse). Based on your synthesis in the discussions section you could also recommend setpoints for the miniCAST to replicate soot types that are found in the ambient. That information is in the manuscript (and in the references) already but is not explicitly stated.

*A large part of this feedback was answered in the previous comments (page 2). This feedback was taken to improve the introduction, i.e. to clarify the objective of the paper, and to improve the writing in Sect. 2 to 5 in order to show the relevance of analysed soot and obtained results.*

*In particular, the following text has been added in the introduction:*

*"In models soot is commonly represented by its BC and BrC components and the evolution of optical properties with atmospheric ageing is parametrized using the properties of freshly emitted aerosols as baseline values (e.g. Samset et al., 2018; Sand et al., 2021). Therefore, knowledge of the spectral optical properties of fresh soot aggregates, despite those representing only the initial stages of the particle's lifetime, is relevant to modeling the impact of soot along its whole lifecycle, from emission to deposition. For BC a commonly accepted value for the MAC is $7.5 \pm 1.2$ $m^2 g^{-1}$ at 550 nm, as proposed by Bond & Bergstrom (2006), further discussed by Bond et al. (2013), and more recently confirmed by Liu et al. (2020b) who evaluated a value of $8.0 \pm 0.7$ $m^2 g^{-1}$ at 550 nm as a representative value for BC-dominated mature fresh soot. Conversely, lower scientific consensus is established for the spectral optical*

*properties of primary combustion BrC (Saleh, 2020). The BC and BrC are co-emitted in different proportions and with different properties for varying fuels and combustion conditions, which reflects differences in particle maturity (e.g., Schnaiter et al., 2006; Mamakos et al., 2013; Török et al., 2018; Cheng et al., 2019; Ess et al., 2021; Malmborg et al., 2021). Higher proportions of BrC are emitted from biomass or low-quality fuels that burn at lower temperatures and fuel-rich conditions, while lower proportions of BrC are emitted from the combustion of fossil fuels at high temperatures in internal combustion engines (Chen and Bond, 2010; Cheng et al., 2019; Corbin et al., 2019; Saleh et al., 2018). The variable BC–BrC mixture in soot, commonly expressed in terms of aerosol's elemental-to-total (EC/TC) or elemental-to-organic (EC/OC) carbon ratio or also as BC-to-organic aerosol BC/OA mass ratio, is accompanied with changes in the soot optical properties, including the intensity of interaction with light (absolute values of MAC/MSC/MEC/SSA) and its spectral dependence, represented by the Absorption and Scattering Ångström Exponents (AAE, SAE) (e.g., Cheng et al., 2019; Ess et al., 2021; Ess and Vasilatou, 2019; Kelesidis et al., 2021). Notably, it is observed that the BrC spectral absorption changes with particle maturity (Saleh et al., 2014) that implies that the contribution of BrC to soot absorption is not linear for changing the maturity of soot. In consequence, as discussed in Saleh (2020), for combustion emissions containing both BC and BrC it can be more convenient to parametrize the effective optical properties of the BC–BrC mixture instead of its single BC and BrC component alone. However, as of today, still a robust relationship linking soot optical properties to particle's composition is missing, which represents a limitation to accurately represent absorbing combustion aerosols in models."*

Concerning the recommendation of a specific miniCAST set point to replicate soot types found in the atmosphere, we would not suggest any specific operation point in the paper. In the manuscript however we mention current knowledge on similarity between miniCAST soot and ambient aerosols. For instance, it is well established that fuel-lean combustion conditions in miniCAST produce soot whose properties best resemble the ones of fresh combustion aerosol from diesel and aircraft emissions as discussed in P6 L165 (now P4 L115-116).

The strength of this work is in the discussions section of the manuscript where the experiments are put into context with previous research and other types of generated aerosols. Before reaching the discussions section, one has to read quite a lot of details about the experiments. I would not mind if the authors could review if everything in sections 2 and 3 is needed in the revised manuscript, keeping in mind that something could be put into the supplements, such as e.g. Fig. 2 and Fig. 6.

Sections 2 and 3 have been revised and shortened as suggested by the reviewer. In particular, the details on data acquisition and analysis for instruments that are less central in the analysis have been moved into Supplementary Information. This concerns: the description of the SP2, ACSM, and TEOM operations and data analysis in moved into Text S2; the details of data acquisition and analyses for the offline filter-based absorption techniques, which is moved into Text S3; Figure 2 has been moved into Fig. S1. The main text has been changed accordingly to reflect these changes. Concerning Fig. 6 (now 5) we decided to keep the figure.

**Specific comments**

P3 L75-78: I don't understand the reasoning here. Light scattering by aerosol particles is mainly due to the size of the particles and their refractive index, whereas light absorption is due to the amount of BC in the aerosol. In the atmosphere, BC is often coated or at least attached to a non-absorbing material. If extinction is dominated by absorption then it means that the particle is

mainly BC or so small in size that it does not scatter light. Why not talk about single scattering albedo (SSA) here instead?

In the effort to simplify and optimise the introduction this sentence was removed. But we agree that the specific sentence was not precise enough and the SSA would be the more common way to express exactly the same concept that is pointed out here.

P4 L103: There is no mention of the size dependence of MAC, MSC, MEC and SSA in the introduction. The primary spherules of the fractal aggregates (shown in Figure 3) are surely in the Rayleigh regime and are therefore volume absorbers (https://doi.org/10.3155/1047-3289.59.9.1028). Light scattering is proportional to the optical size of the particles; see the size parameter x in Fig. 1 of https://doi.org/10.3155/1047-3289.59.9.1028. I suggest to also include the SAE when reporting MSC in the article as the SAE is a measure of the optical size of the particles. This information is included in Table 2, but not in the text in the results/discussion section to the degree that it would become clear that the MSC is indeed largely due to the size of the particles.

Introduction was significantly modified following the other comments by the reviewer so to point out more clearly at the main objective of investigating BC and BrC mixtures in soot and the impact on optical properties. Therefore no specific mention on the dependence on size was made. As pointed out by the reviewer, the SAE and its size dependence is not used a lot in this work but this is the result of the fact that it, in particular under consideration of the uncertainty of the SAE, does not vary or evolve significantly throughout an experiment. As for example visible in Figure 1 (panel d) and pointed out in the text. This is despite the constant particle growth due to coagulation. This growth can be found to result in a consistent increase in SAE, that based on precision and uncertainty however cannot be considered significant. Since this consistent change in size allows us to determine the SAE at different CMD values it can be noted that independent of the CMD the SAE of the aerosols differs systematically between the CSs. This means that if the populations are similar in size the SAE remains different and thus the main cause for difference in the SAE is likely to be associated to the composition / refractive index. The phrasing (e.g. P14 L399-405) was adjusted in an effort to point this out more clearly.

P4 L111: What single soot particles? The sentence is too long and raises more questions than it answers. A lot of references to BC-like soot particles and an ongoing discussion but I would urge the authors to provide your thoughts and summary on this issue as few readers know all these publications by heart and immediately know what you mean.

In the effort to simplify and optimise the introduction the specific sentence was removed but the feedback was taken into account for the reworking of this section.

P7 L216-220: Where is this ACSM data used? This data could be useful when trying to understand the CS5 experiment in greater detail. On P15

L442-445 there is a mention of this data, but the reliability of the results is put into question.

It is discussed in P15 442-447 (now P12 329-333) that the CS OC is not vaporizing either due to bounce-off or due to the properties of the OC – thus despite using the ACSM correctly for that purpose, it doesn't allow for a quantification or characterisation of the OC components. The lack of measurement on OC fraction however provides additional information on the aerosol properties i.e. that also the OC fraction is likely thermally stable and includes large parts

of pyrolytic carbon. These observations provide interesting information on the nature of the OC and that large parts of the OC fraction represent low maturity soot (low amounts of graphitisation).

P11L332-335: Please revisit this sentence as it is a bit unclear what is meant here.

Was rephrased in hope to improve the understanding of the message.

P14 L407: It is not clear to me why the distributions of Fig. 4 should be gaussian. If the aerosols in the chamber is not monomodal, or if the particle size distribution changes due to e.g. coagulation, I don't see reason for why a gaussian fit is the best matrix for the cross section statistics. For some experiments it works suprisingly well, but for some I'm sure that e.g. a mean value would make more sense than to report the peak of the fit. Please, consider to at least report both. The figure itself is informative and supports the text and overall message that the experiments were successful and well constrained.

The observation is that for experiments with high number of repeatability the data follows a Gaussian while for CS with low number of experiments stronger deviations are found. Thus it can be assumed that the deviation away from a Gaussian is caused by low number of repetition and thus is a statistical artefact. Minimal differences between the Gaussian fit –considering the uncertainties of the parameters- and the statistical mean exist thus there is no additional information to be gained by discussing mean values that assume the same underlying statistics.

P15 L435-438: Please try and make this sentence more clear.

The sentence was adjusted to try to clarify the meaning.

P15 L439-447: As I suggested before, I would like to see the CS5 experiment separated from the CS1-4 experiments since it dilutes the discussion of CS1-4 too much.

CS5 is needed as the end point of the mixing curve and it needs to be evaluated in-line with the properties from CS1 to CS4 and in particular of the OC – fraction of these CSs. This is a key discussion for the continuity in the EC/TC-Optics relation that is key in the work, thus we consider it necessary to be discussed together with the other CS points. But based on the review the discussion was adjusted and an effort was made to reduce any imbalances between the CSs.

P21 L578: What is meant with "over particle lifetime"? That OC particles in the Aitken mode coagulate with the larger soot particles in the chamber?

Formulation was adjusted, but yes the smaller particles coagulate with the larger particles during the time of which the aerosol is in suspension inside the chamber.

P22 L606-609: Can the authors recommend setpoints for the miniCAST to reproduce some of these aerosol types that are found in the ambient air? This would be valuable for the scientific community.

As the review points out, the properties of ambient aerosol are variable and can change from fresh combustion aerosol. So there is no set-point that allows to generally predict atmospheric aerosol in particular as the generated soot presents the properties of fresh combustion aerosol. Therefore we do not consider that miniCAST set points reproduce any specific soot type, rather

the miniCAST is a laboratory standard tool to generate under controlled conditions soot with different physico-chemical properties. Nonetheless, in the manuscript we mention current knowledge on similarity between miniCAST soot and ambient aerosols. For instance, it is well established that fuel-lean combustion conditions in miniCAST produce soot whose properties best resemble the ones of fresh combustion aerosol from diesel and aircraft emissions as discussed in P6 L165 (now P4 L115-116). The studied range here encompass the range of possible soot EC/TC values in the atmosphere going from fuel-lean to fuel-rich condition, as the main objective is to get a systematic understanding of impact of varying maturity and BC/BrC/OC-mixing on the particles' optical properties.

P22 L610: Remove "Relating the..." from the heading.

The text was adjusted.

P23 L615-617: Please split this one sentence into more sentences to make the message clear. Now the message is too convoluted. Isn't the spectral dependence of absorption (please use AAE) depending on the mixture of fuel, oxidation air and N2 and not the "absolute" value of these?

Following reviewers' comment, the overall sentence was simplified and split in two. It now reads as:

"The spectral dependence of the absorption, expressed as the AAE, varies with the changing fuel, oxidation air, and mixing $N_2$ flows. The change correlates well with the combustion dependent change in the size, the relative presence of the organic fraction, and thus also the presence of the potentially less mature soot fraction."

"Absolute" value was removed as theoretically both is right. While the composition is found to be mainly dependent on the relative values of C and O in the combustion process (e.g. Schnaiter et al., 2006), the absolute values impacts soot concentration and properties such as size in the miniCAST (e.g. Moore et al., 2014).

P24 Figure 7: I think panels (a) and (b) in Figure 7 are very informative and a great addition to literature and potentially useful for modelers. Having said that, I don't see the point of panels (c) and (d). The SAE and MSC are mainly a function of particle size and has little to do with the OC/EC ratio. Surely the OC/EC ratio will change the refractive index (and density) of the partilces and thus also impact the SAE and MSC, but SAE will still be dictated by the size of the particles (and MSC by the density and size of the particles).

P25 L697-P26 L698: "the SAE suggest a potential similar relation between composition and spectral dependence of scattering as observed for the absorption" This is because the particle size changed between the experiments CS1-CS5 (CMD in Table 1). The spectral dependence of scattering (please use SAE) depends primarily on particle size and not the OC/EC ratio.

In regards to the previous two points: As mentioned above, during the experiments constant particle growth is observed due to coagulation. This growth can be found to result in a consistent increase in SAE, that based on precision and uncertainty cannot be however considered significant. It can be noted that, independently of the CMD, the SAE of the aerosols differs systematically between the CSs. i.e. if the soot aerosol populations are similar in size the SAE remains different and thus the main cause for difference in the SAE were associated to the

refractive index and composition of the particles. Therefore, we prefer to keep the panels c) and d) in Figure 7 (new Figure 6) to provide analysis of the possible relationship between MSC and SAE with CS point/composition so that the analysis is complete.

P26 L710: MSC is primarily a function of particle size and particle density, and not EC/TC content.

Considering the precision of measurements, it is not possible to evidence any differences in the MSC between the five CS, even for the much smaller CS5. As a matter of fact, our experiments do not allow to put in evidence significant differences in the MSC related to the size and chemical composition between the different CS points. Considering the limited amount of data in the literature as well, we conclude this aspect could deserve more investigation.

**Technical corrections (P = page, L = line)**

P2 L41: "...during the combustion ..." remove "the" – Was removed

P2 L45: "laser incandescence" add induced – Was added

P4 L131: remove "realism" - Was removed

P5 L134: remove "tries" - Was removed

P8 L258: change "cast soot" to CS. Remove "around" - Was removed

P11 L342: Please change to "190 to 640 nm wavelength range" – Was changed

P21 L 564: BrCACN is not defined yet. – Defined in P11 (305) but also adjusted here to remind of the synonym.

P22 L581: wavelength dependency should be AAE? – Not changed as a higher wavelength dependence of the absorption implies a higher AAE as the latter is the descriptive parameter for the other.

P22 L583: Change "could" to can. – Was changed

P22 L588: "Seems to" is rather vague. – was removed

P22 L599-600 Please rephrase "such a less" - Was changed

P22 L601: Please remove "As a matter of fact". - Removed

P23 L 628: Please remove "in particular". – Was kept

P23 L 631: change "transformed" to calculated. - Was changed to adjusted

Figure 6. Label the panels (a) and (b). – Was changed

Figure 7. Schnaiter et al. 2003 and Ess et al. 2019 are too similar in color, and so is Kumar et al. 2018.

Yes, both studies are reported in blue, but there are significant differences between the two colours. For Kumar et al. (2018) the same colour was used (green) as data belong to the same paper but separated in description to show that this work supplied both an BC and POA value which are on the opposite side of the spectrum so easily distinguishable).

Supplement:

P2 L14: Why is this an alternative? I suggest just saying that it is the same as Fig. 4, but using the 630 nm wavelength. - Was adjusted

P3 Figure S3: Same as for Figure 7 panes (c) and (d) do not make sense to me so I would remove them. You could try and plot SAE vs MSC for your experiment. Schnaiter et al. 2003 and Ess et al. 2019 are very similar in color, and so is Kumar et al. 2018.

 See above.

P4 Figure S5: The blue wavelengths seems to match well but red (630 nm) shows higher extinction although both instruments should show the same value. Is this a calibration issue or can it be because they were not connected "simultaneously". I'm not sure what it means that they were not connected simultaneously. The clock of the logging software should have been the same. Or is it a delay in the instruments or in the sampling lines.

P5 Table 1: It would be very interesting to see the results of the chemically aged experiments in this manuscript. Especially how the optical properties and the TEM images of the CS aerosols from those experiments. If the soot cores collapsed during those experiments, that would provide much sought after information about how the CS aerosols you have generated would act if subject to more ambient like conditions, which the manuscript now lacks. The scientific impact of this paper would substantially improve by including the aged CS aerosol experiments.

Differences between the instruments are considered to be associated to calibration and uncertainty due to interpolation (the 630 nm - $b_{scat}$ value from the nephelometer is interpolated from 550 and 700 nm). Nonetheless, when stable conditions are achieved in the chamber, the two measurements tend to agree within their own uncertainties. The instruments were not connected simultaneously as they are on two different sampling lines connected to the chamber, which causes the apparent differences. Results are only produced and compared for periods where all instrument was continuously connected.

P6 Table 2: You might want to include the size range of the SMPS in the manuscript for easier reading. - It can be found on P7 204 (now P5 L152)

P8 change "Pattenuation" to attenuation. – Was changed

References

https://doi.org/10.3155/1047-3289.59.9.1028

https://doi.org/10.1029/2005JD006389

https://doi.org/10.1029/1998JD100069

https://doi.org/10.1016/j.carbon.2024.119197

https://doi.org/10.1038/ngeo2901

**Reviewer No. 2:**

The authors generated soot aerosols with a miniCAST burner with different particle sizes, maturity, and composition, i.e. elemental to total carbon ratio (EC/TC). The particles were injected into a large aerosol chamber known as CESAM, where they were suspended for a few hours to investigate the evolution of the particle physico-chemical and optical properties. In this study, the authors only performed measurements under dry, dark conditions in an O2+N2 atmosphere.

The manuscript is well structured and written in a clear language. The authors have put considerable experimental effort, carrying out measurements with an array of different instruments. The figures are informative and the acquired data are compared to those obtained by past studies. The authors seem to be very knowledgeable and familiar with the topic, which is a major advantage of this study.

The main drawback of this manuscript is, in my opinion, that this study (despite its length) is very limited in terms of test aerosols and ageing conditions in the CESAM chamber. All five test aerosols were generated with a miniCAST under fuel-lean or fuel-rich conditions and were not processed any further (e.g. ageing with ozone, controlled coating with primary or secondary organics, coating with inorganic substances, such as sulphuric acid etc.). In addition, all measurements were performed under dry and dark conditions. All test aerosols had a fractal-like morphology although the majority of combustion particles in ambient air (apart from freshly emitted soot) will have a much more compact structure, which will in turn affect their optical properties.

Questions:

- Could the authors explain the narrow selection of test aerosols and experimental conditions used in this study?

The objective of this work was to characterise the variability of the spectral optical properties of fresh combustion aerosol due to varying combustion conditions and the resulting change in soot maturity and composition. The current state of knowledge as indicated in the introduction is that BC/BrC – EC/OC- ratios of fresh combustion aerosol differ dependent on the combustion process and thus optical properties of the particles vary with a significant amount of works observing this and providing data. However, only a few works have started to systematically investigate the variability of the EC/OC evolution and optical properties.

The number of soot test aerosols investigated in this work was defined based on two criteria: further prove the systematic change in composition with soot maturity (or combustion condition) independent of a specific soot generator, and test a large range of conditions from fuel-lean to fuel-rich combustion. The amount of data already available in the literature enabled us to show that our results are comparable with other works, particularly for fuel-lean conditions, and extended our observations to provide a more generalized relation of soot optical properties versus composition. Therefore, with the idea of combining our data with the literature, and considering the experimental effort of performing long experiments in a simulation chamber, we did not consider necessary to test a much wider range of conditions than those presented in the manuscript.

Is the primary organic matter generated by the miniCAST representative of the primary organic substances generated by the various combustion sources, such as vehicle or aircraft engines?

This is an important question and surely a question that remains open from the results of this work. The absorbing OC is considered to be largely immature soot and as such it can also be found in other combustion sources. The fact that the few diesel observations added in Figure 7 follow the same optical properties and composition trends as the miniCAST soot suggests that the BrC component has similar properties. However, this work does not have the data to prove this definitively and extensively for large varieties of sources. More advanced chemical analysis identifying and comparing the chromophores for the miniCAST vs vehicles and aircraft engines should be performed to validate this hypothesis.

- To which extent are the results, especially the optical properties, presented in this study realistic of ambient aerosols, taking into account that most soot particles in the atmosphere have a collapsed structure and a coating (which could either be transparent or light absorbing). In that respect, under which conditions can the results presented in this study feed directly into radiative transfer models?

The shown spectral optical properties are limited to the fresh combustion aerosol. In the atmosphere the soot aerosol are modified and undergo physical and chemical ageing that also alters the soot optical properties, and thus ambient combustion aerosol can differ from the properties shown in this work. Currently this change in optical properties and the variability in optical properties is considered in models by adjusting fresh soot values (such as provided by this work) to the processing the particles undergo via parameters such as the Enhancement Factor.

Therefore, the spectral optical properties of fresh soot, despite them being relevant only for a brief residence time in the atmosphere, are key to represent soot in models and follow their changes. Therefore, our results can be directly used in radiative transfer schemes and can serve as basis for representing the initial soot optical properties, as they are emitted in the atmosphere. The relathion against EC/TC in particular can allow to account for the diversity of fresh soot properties for changing particles' combustion condition and maturity.

In order to clarify this important aspect, the introduction and the conclusion have been modified. In particular, the text has been changed to clarify why the chosen conditions were used and how the work can contribute to improve the understanding of atmospheric combustion aerosol.

- In my opinion, miniCAST burners can only generate realistic aerosols under fuel-lean conditions, when the % EC/TC mass fraction is high. These particles can simulate the properties of e.g. diesel soot. In response to the question raised by Reviewer #1, namely whether the authors can recommend miniCAST setpoints to replicate different soot types found in the ambient air, I don't think this is possible. I would recommend that the soot particles from the miniCAST generator be denuded and then coated in a controlled manner with primary/secondary organic or inorganic substances, see for instance:
- Kalbermatter et al. doi.org/10.5194/amt-15-561-2022
- Pagels et al. https://doi.org/10.1080/02786820902810685
- Khalizov et al. https://doi.org/10.1021/jp807531n

Especially Dr. Khalizov has published numerous papers in this field.

Yes, the opinion that the soot from fuel-lean combustion in generally representative of realistic aerosol is in agreement with the literature as also summarized in this work. As indicated in the manuscript soot similar to the fuel-rich combustion are possible but rather specific. The studied combustion (fuel-lean to fuel-rich) range here encompass the likely values in the atmosphere however the focus of this work is on and the systematic understanding of the impact of varying maturity (and BC/BrC/OC-mixing) on the soot optical properties. Using the fuel-rich CS (highly immature soot) extends the EC/TC range and range of observations which in turn allows to retrieve the systematic relationship proposed.

As indicated in the answer to the previous question the understanding of coating and collapse structures is key to represent ambient soot aerosol and its variability. And works given above by researchers such as Dr. Khalizov (mentioned by the reviewer) are of utmost importance for this. A large part of these works however quantifies the change in optical properties as a relative value (enhancement factor E or EF) relating pre-ageing values to values of aged aerosol. This relative values are also used to apply the evolution of properties in models. Thus, the results from these previous papers further stress the importance to understand the fresh soot properties and constrain their likely values as good as possible in order to have a solid basis for both the ability to study and to model atmospheric processing.

I would recommend that the authors provide an outlook with suggestions on how to overcome the limitations of the present study.

Based on this feedback and the questions above the manuscript was adapted in particular the introduction to clarify this aspect.

- Page 3-Lines 95-97: The instrument PTAAM-2λ can serve as a reference instrument (even traceable primary standard) for aerosol light absorption

Drinovec et al. https://doi.org/10.5194/amt-15-3805-2022

From the given sources, yes, but the questioned instrument was not well established in both literature and measurement reports during the time of the performed experiments, i.e. early 2021 (experiments) vs 2022 (publishing date of the given reference). The low amount publications on this instrument at this time therefore make it difficult to consider it as a reference measurement for these specific experiment. The authors would be pleased to consider it for further investigations.

Minor points:

Abstract and main text: The term "propane soot" can be misleading. I would recommend to simply explain that the miniCAST was operated with propane as fuel.

The abstract was modified to account for this suggestion.

Page 2 / Line 46: Replace "specie" by "species" – Was adjusted

Page 1/Lines 31-33: Please reformulate the sentence, especially the following part " …representing the spectral absorption of soot with varying maturity to lie in an optical continuum" – Was adjusted as:

"A combination of our results for soot from propane combustion with literature data for laboratory flame soot from diverse fuels supports a generalized exponential relationship between particle EC/TC and its MAC and AAE values, which represents the optical continuum of spectral absorption for soot with varying maturity."

Please expand the acronyms/abbreviations "UV" and "UV-Vis" – The text was adjusted where relevant

Please add a space between the numerical value and the units, e.g. 600 °C. Note that % is also unit and should be separated by a space. – Was adjusted

Page 4- Lines 131-132 & Page 7-Lines 222-224: These sentences sound odd "Taking advantage of the realism of aerosol suspension in a large chamber…". "To note that due to the TEOM sensitivity to even slight pressure changes and potential consequential data instability, its data are mainly used to validate mass calculations from complimentary approaches described below". Consider reformulating. – First sentence was adjusted. The second sentence was removed as part of an effort to optimize/shortening the manuscript.

Page 8 -Line 254: Consider replacing "resulted often" by "often resulted"– Was adjusted

I would recommend to harmonise notation, e.g. that all measured quantities (D, ρ, b etc.) appear in italic. – homogenised throughout the paper

Figures: I would recommend that the units be put in parentheses instead of square brackets. Square brackets are used as operators to extract the units from a measured quantity, e.g. [m]=kg (the unit of mass is the kilogram). - Both notations can be found throughout literature thus we will remain with the given one.

Page 15-Line 441: "… that we therefore assume to be equal to 0.0" – *adjusted to 0*

Page 16/Line 468: "has to be noted, that the number of particles smaller…". Please remove the comma. – Was adjusted

Page 18/Line 525: Consider changing "For all generated soots the MAC is dominating over…" to " For all generated soot aerosol, the MAC is dominating over…". The word "soots" also appears on Page 25/Line 690 and should be corrected to "soot particles/aerosols".

Was adjusted

Caption of Table 2: "Provided uncertainties represent the combination of statistical and measurement uncertainties…". Could the authors please explain what is meant by "measurement uncertainties"? Do these uncertainties originate from the calibration of the instruments against a reference method?

Throughout the text: Please make sure you state the coverage factor and confidence interval whenever you report measurement uncertainties and explain what these uncertainties represent.

Replying to the previous two comments. Measurement uncertainties are the uncertainties related to the retrieved/measured parameters. Also if used to calculate a secondary value such as the mass cross sections or the Angstrom exponents for which the uncertainties are propagated from measured values following typical error propagation. The origin of the individual measurement uncertainties are described in Table S2. In table S2 and this work it is generally indicated if uncertainties (described in table S2 (generally equivalent to 2 standard deviations) or propagated from these values following typical error propagation) or standard deviations (=single standard deviations i.e. coverage factor =1) are used.

---

## Author Response (AR2)

Comment on "**Spectral optical properties of soot: laboratory investigation of propane flame particles and their link to composition**" by Johannes Heuser, Claudia Di Biagio, Jerome Yon, Mathieu Cazaunau, Antonin Bergé, Edouard Pangui, Marco Zanatta, Laura Renzi, Angela Marinoni, Satoshi Inomata, Chenjie Yu, Vera Bernardoni, Servanne Chevaillier, Daniel Ferry, Paolo Laj, Michel Maillé, Dario Massabò, Federico Mazzei, Gael Noyalet, Hiroshi Tanimoto, Brice Temime-Roussel, Roberta Vecchi, Virginia Vernocchi, Paola Formenti, Bénédicte Picquet-Varrault, and Jean-François Doussin

**General comments:**
The manuscript has in my opinion been substantially improved and is now much easier to follow and more clear. The authors have made significant improvements in clarifying the scope and limitation of their work to a satisfactory level which warrants publication in the journal. However, I have requested minor correction to the revised manuscript, since some of the criticism has not been properly addressed. I realize that this might be in part due to me not putting my statements as clearly and bluntly as needed. Below I raise my concerns of the remaining issue with the revised manuscript.

**Original referee comment:** "P24 Figure 7: I think panels (a) and (b) in Figure 7 are very informative and a great addition to literature and potentially useful for modelers. Having said that, I don't see the point of panels (c) and (d). The SAE and MSC are mainly a function of particle size and has little to do with the OC/EC ratio. Surely the OC/EC ratio will change the refractive index (and density) of the particles and thus also impact the SAE and MSC, but SAE will still be dictated by the size of the particles (and MSC by the density and size of the particles).
P25 L697-P26 L698: "the SAE suggest a potential similar relation between composition and spectral dependence of scattering as observed for the absorption" This is because the particle size changed between the experiments CS1-CS5 (CMD in Table 1). The spectral dependence of scattering (please use SAE) depends primarily on particle size and not the OC/EC ratio."

**Authors' reply:** In regards to the previous two points: As mentioned above, during the experiments constant particle growth is observed due to coagulation. This growth can be found to result in a consistent increasedecrease in SAE, that based on precision and uncertainty cannot be however considered significant. It can be noted that, independently of the CMD, the SAE of the aerosols differs systematically between the CSs. i.e. if the soot aerosol populations are similar in size the SAE remains different and thus the main cause for difference in the SAE were associated to the 9 refractive index and composition of the particles. Therefore, we prefer to keep the panels c) and d) in Figure 7 (new Figure 6) to provide analysis of the possible relationship between MSC and SAE with CS point/composition so that the analysis is complete.

**Original referee comment:** P26 L710: MSC is primarily a function of particle size and particle density, and not EC/TC content.
**Authors' reply:** Considering the precision of measurements, it is not possible to evidence any differences in the MSC between the five CS, even for the much smaller CS5. As a matter of fact, our experiments do not allow to put in evidence significant differences in the MSC related to the size and chemical composition between the different CS points. Considering the limited amount of data in the literature as well, we conclude this aspect could deserve more investigation.

**Referee comment on the revised manuscript**:
The listed discussion cited above does not address the concerns I raised to a satisfactory level and my opinion on panels c and d (new Figure 6) remains. The authors have tried to justify keeping them in their reply, but the reply is not convincing, nor has it lead to any significant change in the manuscript to highlight that SAE depends to a much greater extent on particle size than the OC/EC ratio, as one would be made to believe based on Figure 6c in the revised manuscript.

**On p. 29 L 791 in the revised manuscript it reads:** "*The lack of literature data for these values however did not allow to acquire enough information for any significant synthesis. However, while the MSC data do not show a clear dependence, the plot for the SAE suggests a potential similar relation between composition and spectral dependence of the scattering, as observed for the absorption.*"
Again, The SAE is mainly a function of particle size and has little to do with the OC/EC ratio, other than that the OC/EC ratio explains under which miniCAST conditions the aerosols were produced

at. To put it bluntly, plotting OC/EC ratio with SAE without considering the size of the generated particles is misleading. Especially with the quoted statement above that claims that there is a similar relationship between "composition and spectral dependence of scattering, as observed for the absorption". This is bluntly put not true. Quoting the Hinds (1999, p. 370) textbook "Light scattering provides an extremely sensitive tool for the measurement of aerosol concentration and particle size.". This can be seen in Figure 1, contrary to what the authors argue in their reply, where the SAE drops as the particles grow through coagulation in the chamber. For atmospheric aerosols, the SAE is in the range of 0 (coarse mode aerosols) to 4 (nucleation mode aerosols & air molecules i.e. Rayleigh scattering regime).

Moreover, light absorption and light scattering are physically two different processes, the latter being, as mentioned, primarily a function of particle size. This is stated in any textbook on the subject and should not be confused with the physical process of light absorption. Furthermore, making a parametrization for modelers for MSC and SAE using the OC/EC ratio can make readers not familiar with aerosol optics adopt the scheme. The results are specific for the miniCAST burner. In their reply, the authors state that

"*It can be noted that, independently of the CMD, the SAE of the aerosols differs systematically between the CSs. i.e. if the soot aerosol populations are similar in size the SAE remains different and thus the main cause for difference in the SAE were associated to the 9 refractive index and composition of the particles.*"

Where in the manuscript is it shown that the CMD and SAE are independent of each other for different CSs? I suggest removing panels c and d from Figure 6 in the revised manuscript.
If the authors are very keen to keep them, move the panels c and d of Figure 6 to the appendix.

**AUTHORS REPLY:**

First, we want to thanks the reviewer for having provided further comments to ameliorate the manuscript. The authors agree that size is the key property for scattering behaviour of aerosols as it is determined by the underlying physics of aerosol light interaction, as it is well established by the reviewer. We acknowledge that the provided plots (Fig. 7 panels c and d) could be misleading as the proposal of a generalised relation for soot could lead to a neglect of the key property of particle size when modelling the scattering properties. We therefore accept the reviewer's suggestion and we move panels c) and d) of Figure 7 in the Supplementary Information.

We additionally modified the text in some specific parts to reflect these changes, as clearly identified in the tracked manuscript. Main changed paragraphs are:

*Abstract*: "In this study, soot aerosols with varying maturity and composition, i.e. elemental-to-total carbon ratio (EC/TC), have been studied systematically in a large simulation chamber to determine their mass absorption, scattering, and extinction cross sections (MAC, MSC, MEC), single scattering albedo (SSA), and Absorption and Scattering Ångström Exponents (AAE, SAE). The MAC, MEC, SSA and AAE show a variability between the different soot with varying EC/TC ratios."

*Paragraph of Sect. 5.2 has been modified as*; "An alternative plot relating the MAC and AAE to the also commonly used EC/OC-ratio can be found in the Supplement (Fig. S4). For completeness, also plots for the MSC and the SAE vs EC/TC and EC/OC are shown in Fig. S5 and S6, despite scattering depends primarily on size and relation to composition has limited significance."

*Conclusions*: "The dependence of the MAC, MSC, MEC, and SSA for the five generated soot aerosols and their spectral variability, represented by the AAE and SAE, on the soot composition were analysed. While the MSC appears independent of the soot type (0.4-0.5 $m^2g^{-1}$ at 550 nm), all other parameters show a variability associated with the soot and its composition. "

Besides changes based on the reviewer's feedback, minor changes to the text have been made to adjust phrasing and correct spelling. These changes do not alter the content of the manuscript and can be followed in the tracked manuscript.